# An invertible, invariant crystal representation for inverse design of solid-state materials using generative deep learning

Hang Xiao [1], Rong Li[2], Xiaoyang Shi[3], Yan Chen[4] ✉, Liangliang Zhu [2,5] ✉, Xi Chen [1] ✉ & Lei Wang [6,7] ✉

The past decade has witnessed rapid progress in deep learning for molecular design, owing to the availability of invertible and invariant representations for molecules such as simplified molecular-input line-entry system (SMILES), which has powered cheminformatics since the late 1980s. However, the design of elemental components and their structural arrangement in solid-state materials to achieve certain desired properties is still a long-standing challenge in physics, chemistry and biology. This is primarily due to, unlike molecular inverse design, the lack of an invertible crystal representation that satisfies translational, rotational, and permutational invariances. To address this issue, we have developed a simplified line-input crystal-encoding system (SLICES), which is a string-based crystal representation that satisfies both invertibility and invariances. The reconstruction routine of SLICES successfully reconstructed 94.95% of over 40,000 structurally and chemically diverse crystal structures, showcasing an unprecedented invertibility. Furthermore, by only encoding compositional and topological data, SLICES guarantees invariances. We demonstrate the application of SLICES in the inverse design of direct narrow-gap semiconductors for optoelectronic applications. As a string-based, invertible, and invariant crystal representation, SLICES shows promise as a useful tool for in silico materials discovery.

The past decade has seen rapid progress of inverse molecular design using generative models (GMs)[1]: de novo design of drugs[2,3], synthetic routes of organic compounds[4,5], as well as generative design of molecular electronics[6,7]. For molecules, there are several invertible and invariant representations such as simplified molecular-input line-entry system (SMILES)[8], International Chemical Identifier (InChI)[9], and molecular-graph[10]. Invertibility means that representations can be reversed or transformed back to their original structures, whereas invariances indicate that representation after rotation, translation, and permutation is mapped to the same structure without these

[1]School of Interdisciplinary Studies, Lingnan University, Tuen Mun, Hong Kong SAR, China. [2]School of Chemical Engineering, Northwest University, Xi'an 710069, China. [3]Department of Environmental and Sustainable Engineering, State University of New York at Albany, Albany, NY 12222, USA. [4]Laboratory for Multiscale Mechanics and Medical Science, SV LAB, School of Aerospace, Xi'an Jiaotong University, Xi'an 710049, China. [5]Shaanxi Institute of Energy and Chemical Engineering, Xi'an 710069, China. [6]National Laboratory of Solid-State Microstructures, School of Physics, Nanjing University, Nanjing 210093, China. [7]Collaborative Innovation Center of Advanced Microstructures, Nanjing University, Nanjing 210093, China. ✉e-mail: yanchen@xjtu.edu.cn; zhu.liangliang@nwu.edu.cn; xichen@ln.edu.hk; leiwang@nju.edu.cn

operations. A representation that satisfies both invertibility and invariances is necessary to enable general and property-driven inverse design using GMs. Unlike molecules, however, applying GMs to inversely design solid-state materials remains a long-standing challenge[11], owing to the lack of an invertible, invariant and periodicity-aware crystal representation that covers the majority of elements across the periodic table.

Several attempts have been made to address this challenge. A 3D image-based representation was first proposed by Noh et al.[12] and later enhanced by many studies including Hoffmann et al.[13], Court et al.[14], and Long et al.[15,16]. However, 3D image-based representations are not rotationally invariant and training 3D data models is computationally expensive. Some previous studies directly use lattice vectors and atomic coordinates for structure representation, but these models are not invariant to Euclidean transformations[17–20]. Crystal graph is an invariant representation that encodes both atomic and bonding information between atoms. It was proposed by Xie and Grossman[21] in the original Crystal Graph Convolutional Neural Networks (CGCNN) work and was utilized in many improved variants of CGCNN[22], which have made major impacts on data-driven material property predictions. However, crystal graph is not invertible, hence it is not suitable for the inverse design of crystals using GMs. Recently, Crystal Diffusion Variational Autoencoder (CDVAE) was proposed by Xie et al.[23] to explore the generation of stable materials. Utilizing the invariant multigraph representation, CDVAE reconstructs the input crystal structure in a diffusion process. Nonetheless, its reconstruction rate for crystal structures in the MP-20 dataset[24] (comprising over 45,000 materials with unit cells containing up to 20 atoms from the Materials Project[25]) is merely 45.43%, demonstrating its limited invertibility. In short, no work has demonstrated an invertible crystal representation that satisfies full invariances.

In this work, we propose a string-based invertible crystal representation that guarantees invariances, simplified line-input crystal-encoding system (SLICES). The reconstruction of crystal structures from SLICES strings involves three steps: (1) initial structure generation with graph theory techniques[26], (2) optimization based on chemically meaningful geometry predicted with modified Geometry Frequency Noncovalent Force Field (GFN-FF)[27], and (3) structural refinement using universal graph deep learning interatomic potential[28]. The reconstruction routine of SLICES considerably outperforms past methods in accurately rebuilding input crystal structures while maintaining invariances. We showcase the applications of SLICES in

designing direct narrow-gap semiconductors for optoelectronic applications via deep generative models. Additionally, SLICES-based inverse design framework significantly outperforms past approaches in generating materials with a desired property.

## Results

### SLICES representation

Graphs provide a natural representation for modeling molecules and crystals. A molecule is represented as an undirected finite graph where nodes and edges represent atoms and bonds, respectively (Fig. 1a). However, crystals cannot be represented as undirected finite graphs since they have infinite periodic 3D structures. To this end, a finite graph representation of the infinite periodic structure, called the quotient graph, was introduced by Chung et al.[29] (Fig. 1b). In quotient graphs, atoms and bonds in the unit cells of crystal structures are represented by nodes and edges, respectively. The quotient graph by itself does not provide a unique representation of a crystal structure. To ensure a one-to-one relationship between crystal structures and their quotient graphs, it is necessary to add labels denoting translational periodicity and directions to the edges of quotient graphs[29].

A SLICES string always begins with symbols of atoms in the unit cell (Fig. 1b), encoding the chemical composition of the corresponding crystal structure. In SLICES, edges are represented explicitly in the form $u\,v\,x\,y\,z$ (Fig. 1b), where $u\,v$ are node indices and $x\,y\,z$ denotes the location of the unit cell to connect to. In essence, edge labels specify the translation vectors needed to connect unit cells. For instance, in Fig. 1b, the edge $e_4$ has the label '0 0 1', indicating that $e_4$ connects node $C_0$ to the copy of $C_1$ shifted one unit along the $c$ axis. Edge labels, which specify the translational periodicity of edges, are the defining feature of SLICES. They enable the construction of suitable initial guess structures derived from graph theory (see Methods section for details). Thus, we constructed the string representation for crystal structures by encoding the atomic symbols, node indices, and the aforementioned edge labels (Fig. 1b). To disambiguate node indices from edge labels in the string representation, we utilize 'o', '+' and '-' to denote '0', '1' and '−1' in edge labels, respectively. This encoding guarantees that '0' and '1' in SLICES refer exclusively to node indices, eliminating potential confusion during model training.

Encoding all atoms within the unit cell in SLICES eliminates the need to incorporate crystal symmetry groups, simplifying the construction rules for SLICES. Although this results in a less compact representation, this trade-off is justified given that state-of-the-art

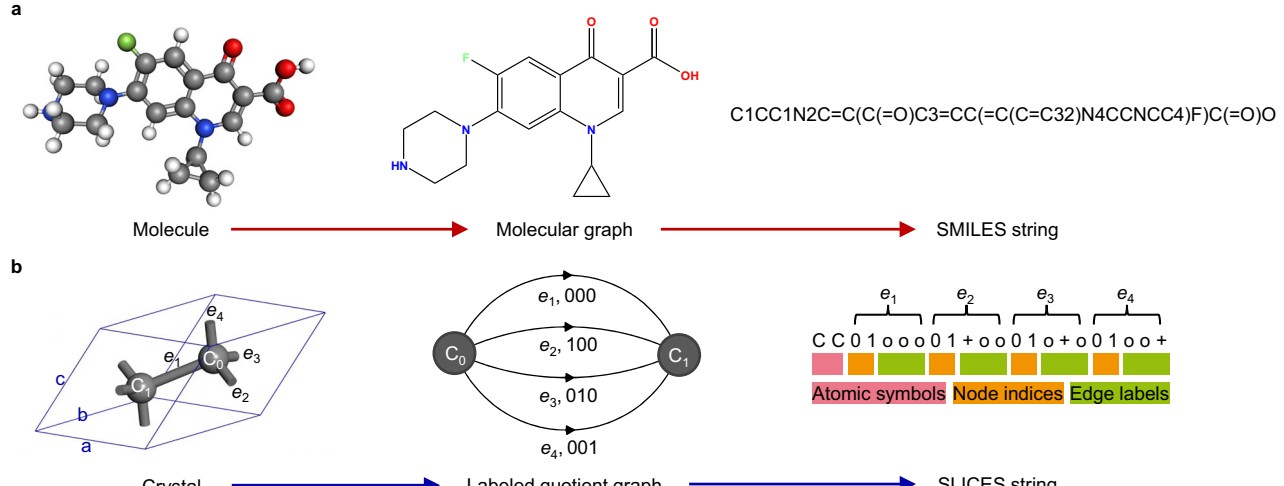

**Fig. 1 | The analogy between simplified molecular-input line-entry system (SMILES) and simplified line-input crystal-encoding system (SLICES). a** The molecular graph serves as an intermediary to translate between molecules and SMILES strings. **b** Likewise, the labeled quotient graph serves as an intermediary to translate between crystal structures and SLICES strings.

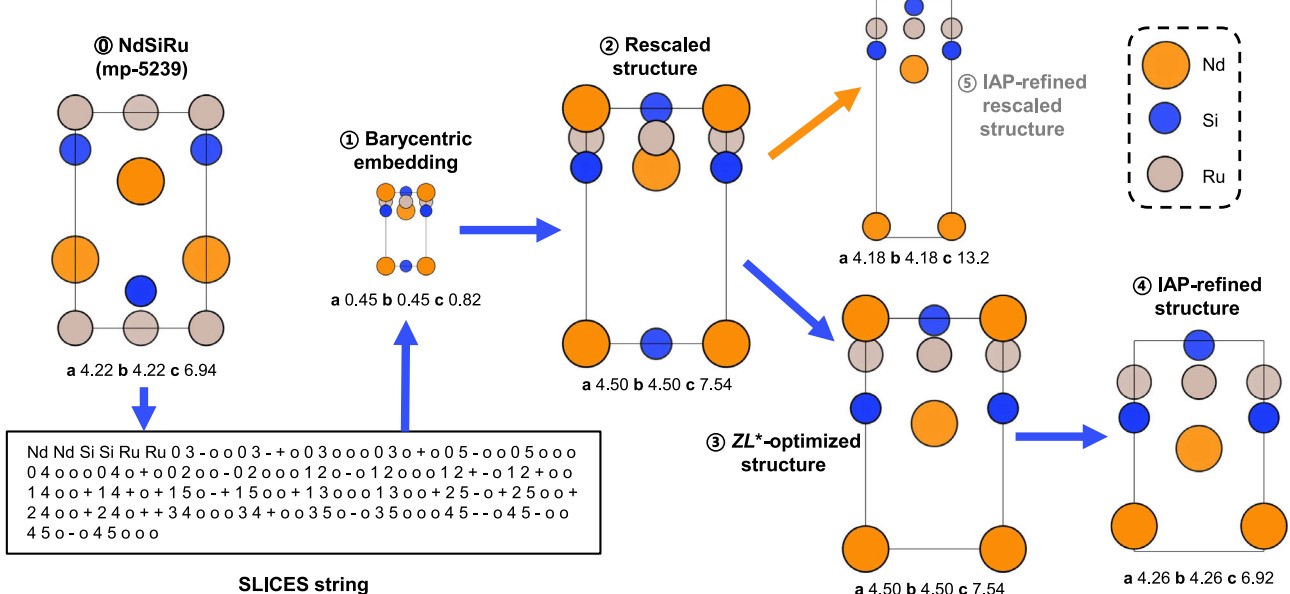

**SLICES string**

**Fig. 2 | Intermediate structures generated during reconstructing the crystal structure of NdSiRu (mp-5239) from its SLICES string.** The original structure of NdSiRu (0) was converted into its SLICES string. The reconstruction process started from generating barycentric embedding (1) using graph theory, followed by rescaling the barycentric embedding to obtain the rescaled structure (2) with modified Geometry Frequency Noncovalent Force Field (GFN-FF). Then, the rescaled structure was optimized based on chemically meaningful geometry predicted with modified GFN-FF to yield the chemically meaningful non-barycentric embedding, denoted as the $ZL^*$-optimized structure (3), followed by structural refinement with the universal interatomic potential for materials based on graph neural networks with three-body interactions (M3GNet IAP) to obtain the IAP-refined structure (4), which matches well with the original structure (0). For comparison, direct refinement of the rescaled structure with M3GNet IAP yielded the IAP-refined rescaled structure (5), which significantly deviates from the original structure (0).

natural language processing (NLP) models excel at handling long sequences, rendering compactness less important. Owing to its simple and clear definition, SLICES could be useful in chemical description and informatics of solid-state materials.

While graph-based representations are more intuitive for crystal structures, string-based representation allows us to take advantage of the extensive and rapidly evolving field of NLP. Based on this consideration, we opted for a string representation over graph-based approaches in this work.

## Encoding crystal structures as SLICES strings

Many methods have been developed to analyze the connectivity of crystal structures following Pauling's pioneering work in 1929 that introduced the concept of bond strength. For instance, minimum distance method, Brunner's method[30], Hoppe's method of effective coordination numbers (EconNN)[31] and crystal near-neighbor method[32] are well-established methods for identifying near-neighbor environments, and have been implemented in the "local_env" module of the Pymatgen[33] package. Among these methods, EconNN[31] offers a relatively good compromise between speed, accuracy and robustness to atomic perturbation[34]. Therefore, analyses of the local chemical environments are performed with EconNN[31] implemented within Pymatgen[33] to define edges (bonds) for SLICES. Specifically, encoding a crystal structure as a SLICES string involves three steps: (1) Parsing the crystal structure from a file (e.g., Crystallographic Information File[35]) into a Structure object using Pymatgen[33]; (2) Constructing a structure graph based on the Structure object using the EconNN[31] algorithm; (3) Extracting the chemical composition, bonding connectivity, and translation vectors from the structure graph to generate the corresponding SLICES string.

## Reconstruction of original crystal structures from SLICES strings

While it is relatively straightforward to create a string-based representation for crystal structures, the difficulty lies in ensuring its invertibility, a key requirement for its implementation in inverse design. In 2003, Delgado-Friedrichs & O'Keeffe[36] developed a graph theory approach to obtain Euclidean embeddings of periodic graph with the maximum acceptable crystal symmetry. This is known as the periodic graph's barycentric embedding, where every node is placed in the center of gravity of its neighbors. Therefore, the barycentric embedding of a labeled quotient graph corresponding to a SLICES string may provide a suitable initial guess structure for reconstructing the original crystal structure.

Note that the barycentric embedding is an embedding with maximum acceptable crystal symmetry. However, crystal structures do not always have maximum acceptable symmetry. In 2011, Eon[26] generalized the concept of barycentric embedding, and proposed a graph-theoretical framework for the systematic generation of non-barycentric embeddings with lower symmetry. Eon's method enables systematic optimization of initial guess structures to satisfy certain geometrical constraints. Based on Eon's method, Boyd and Woo[37] developed a topology-based Metal–organic framework (MOF) constructor called ToBasCCo.

Inspired by these works, we developed a three-step reconstruction scheme from SLICES strings to crystal structures. This reconstruction scheme will be denoted as 'SLI2Cry' throughout the remainder of this text. An example of the reconstruction process for NdSiRu (mp-5239 in Materials Project database) is shown in Fig. 2.

(I) Initial guess structure generation using Eon's topology-based method[26].

The SLICES string was converted into its corresponding labeled quotient graph $G(V, E)$ with $n$ nodes and $m$ edges. $v_i = (s_i) \in V$ is the node $i$ with atomic symbol $s_i$, and $e_j = (u_j, v_j, x_j, y_j, z_j) \in E$ is the edge $j$ connecting node $u_j$ in the original unit cell to node $v_j$ in the cell shifted by $x_j\mathbf{a} + y_j\mathbf{b} + z_j\mathbf{c}$, where $\mathbf{a}, \mathbf{b}, \mathbf{c}$ are lattice basis vectors.

Based on Eon's method[26], if $e_j$ is given, metric tensor $Z$ and edge vectors represented in fractional coordinates of the unit cell, $\Omega$, can be

calculated. $Z$ and $\Omega$ have following forms:

$$Z = \begin{pmatrix} \mathbf{a} \cdot \mathbf{a} & \mathbf{a} \cdot \mathbf{b} & \mathbf{a} \cdot \mathbf{c} \\ \mathbf{b} \cdot \mathbf{a} & \mathbf{b} \cdot \mathbf{b} & \mathbf{b} \cdot \mathbf{c} \\ \mathbf{c} \cdot \mathbf{a} & \mathbf{c} \cdot \mathbf{b} & \mathbf{c} \cdot \mathbf{c} \end{pmatrix}, \Omega(L^*) = \begin{pmatrix} \vec{\mathbf{e}}_1 \\ \vec{\mathbf{e}}_2 \\ \cdots \\ \vec{\mathbf{e}}_m \end{pmatrix} \qquad (1)$$

where $Z$ consists of dot products of the lattice basis vectors, and $\Omega$ is a function of co-lattice vectors, $L^*$ (see Methods for details). Eon[26] showed that co-lattice vectors $L^*$ of an embedding quantify its total deviation from its original barycentric embedding. When $L^*$ is a zero matrix, the combination of $Z$ and $\Omega(0)$ defines a barycentric embedding of $G$ in Cartesian coordinates. Otherwise, the combination of $Z$ and $\Omega(L^*)$ defines non-barycentric Cartesian embeddings of $G$. The details of constructing barycentric and non-barycentric embeddings from $e_j$ are provided in Methods section. The barycentric embedding generated from SLICES of NdSiRu is shown in Fig. 2 as structure 1.

(II) Optimization of the non-barycentric embedding based on chemically meaningful geometry predicted with modified GFN-FF[27].

In step (I), the barycentric embedding was constructed using a purely mathematical approach, without considering the chemical information of nodes ($s_i$). As a result, the lengths of lattice basis vectors are not chemically meaningful. For example, the barycentric embedding of NdSiRu has very small lattice parameters $a$, $b$ and $c$ (Fig. 2). Therefore, the chemical information in SLICES should be utilized to find a chemically sensible version of the barycentric embedding.

In 2020, Spicher and Grimme[27] proposed a universal force field, GFN-FF, capable of effectively modeling a diverse set of systems, ranging from organic molecules to inorganic crystals for elements up to atomic number 86. GFN-FF accepts Cartesian coordinates and elemental composition as input, and subsequently generates a covalent topology encoded in a neighbor list, based on inter-atomic distance criteria. From this topology, GFN-FF calculates a set of topology-based electronegativity equilibrium (EEQ) charges $\mathbf{q_t}$. Subsequently, all potential energy terms can be constructed primarily based on the neighbor list and $\mathbf{q_t}$. The topological nature of GFN-FF allows us to develop a modified GFN-FF, capable of estimating bond lengths and angles from SLICES. Specifically, we modified GFN-FF to take the neighbor list constructed based on $s_i$ and $e_j$ as input, and to output equilibrium bond lengths $l_j^{GFN-FF}$ and equilibrium bond angles $\theta_{jk}^{GFN-FF}$. The details of how we modified GFN-FF are provided in Methods. In summary, this modified GFN-FF is capable of transforming composition and connectivity into chemically sensible geometry.

The equilibrium bond lengths $l_j^{GFN-FF}$ predicted by modified GFN-FF was used to rescale the barycentric embedding constructed in step (I). For brevity, we denote this rescaled barycentric embedding of $G$ in Cartesian coordinates as the rescaled structure. The rescaled structure of NdSiRu, denoted as structure 2 in Fig. 2, exhibits significantly improved lattice parameters.

To compare bond lengths and bond angles of a specific non-barycentric embedding with $l_j^{GFN-FF}$ and $\theta_{jk}^{GFN-FF}$, an inner product matrix $g$ of edge vectors was constructed as follows,

$$g = \Omega\left(L^*\right) \cdot Z \cdot \left(\Omega\left(L^*\right)\right)^t \qquad (2)$$

whose diagonal elements represent squared lengths of the edges, $g_{jj} = (l_j)^2$, and off-diagonal elements contain angular components between edge $j$ and $k$, $g_{jk} = l_j l_k \cos\theta_{jk}$. A target matrix $T$ was constructed with $T_{jj} = (l_j^{GFN-FF})^2$ and $T_{jk} = l_j^{GFN-FF} l_k^{GFN-FF} \cos\theta_{jk}^{GFN-FF}$.

An objective function was defined as follows,

$$O = \sum (T_{jk} - g_{jk})^2 \qquad (3)$$

which is the sum of squared differences between the target matrix and the inner product of edge vectors in a specific non-barycentric embedding. By manipulating $Z$ and $L^*$ to minimize $O$, the chemically meaningful non-barycentric embedding was obtained. For brevity, we denote this chemically meaningful non-barycentric embedding of $G$ as the $ZL^*$-optimized structure. The $ZL^*$-optimized structure of NdSiRu is shown in Fig. 2 as structure 3, of which bond lengths are much closer to the values of the original structure (structure 0 in Fig. 2).

(III) Structural refinement of the $ZL^*$-optimized structure with M3GNet IAP[28]

Although GFN-FF is capable of modeling a wide range of systems, its discontinuous potential energy surface makes it unsuitable for optimizing crystal structures directly. Recently, Chen and Ong[28] developed a universal interatomic potential for materials based on graph neural networks with three-body interactions (M3GNet IAP), which enables high fidelity structural optimization for materials of diverse chemistries. To this end, the $ZL^*$-optimized structure obtained in step (II) was further refined with M3GNet IAP, resulting in the IAP-refined structure (structure 4 in Fig. 2) that matches well with the original structure (structure 0 in Fig. 2). However, direct refinement of the rescaled structure from step (II) with M3GNet IAP resulted in the IAP-refined rescaled structure (structure 5 in Fig. 2) that substantially deviates from the original structure, highlighting the critical role of $ZL^*$-optimization targeting GFN-FF geometry (step II) in the success of SLI2Cry.

## Benchmark on crystal structure reconstruction

The reconstruction performance of SLI2Cry is evaluated by the similarity between the reconstructed and original crystal structures. To evaluate the similarity, we utilized the StructureMatcher function of Pymatgen[33] v.2022.11.7 (Supplementary Note 1). The reconstructed and original crystal structures are deemed similar if they satisfy the matching criteria of StructureMatcher. In particular, two sets of match settings for StructureMatcher were used. Loose: a fractional length tolerance of 0.3, a site tolerance of 0.5, and an angle tolerance of 10°; Strict: a fractional length tolerance of 0.2, a site tolerance of 0.3, and an angle tolerance of 5°. The loose/strict match rate for a dataset is defined as the percentage of reconstructed crystal structures that meet the corresponding loose/strict matching criteria when compared to the original structures in the dataset. The MP-20 dataset[24] curated by Xie et al.[23] was selected in our benchmark. It contains 45,229 structurally and chemically diverse crystal structures, including most experimentally known crystals with 1- 20 atoms in the unit cell. 89 elements from the periodic table are covered in this dataset.

SLI2Cry is universally applicable across datasets without the need of training. This is attributed to SLI2Cry's rule-based steps (I) and (II) that require no training, along with the pre-trained, transferable interatomic potential employed in step (III). Therefore, all data points in MP-20 dataset can be used as testing data. It is noteworthy that the SLICES representation can cover all elements of the periodic table. However, the modified GFN-FF potential employed in step (II) limits the applicability of SLI2Cry to crystal structures containing atoms with atomic numbers up to 86. This restriction leaves us with 42985 crystals (95.04%) in the filtered MP-20. Additionally, while Eon's method[26], applied in step (I), is applicable for 3D crystal structures, it fails for low-dimensional (0D, 1D, or 2D) structures such as molecular or layered crystals due to fragmented quotient graphs. A potential solution is a hierarchical graph approach treating low-dimensional structural units as nodes in quotient graphs, which is planned for future work. Consequently, we eliminated crystals with low-dimensional structural units, bringing the filtered MP-20 down to 40,330 crystals (89.17%).

**Table 1 | Reconstruction performance of SLI2Cry for the filtered MP-20 dataset (40,330 crystals)**

| Setting | Match rate (%) | | | |
|---|---|---|---|---|
| | Rescaled structure | $ZL^*$-optimized structure | IAP-refined structure | IAP-refined rescaled structure |
| Strict | 77.87 | 84.57 | 92.55 | 86.83 |
| Loose | 91.36 | 94.05 | 94.95 | 90.58 |

**Table 2 | Reconstruction performance for the MP-20 dataset (45,229 crystals) under the loose setting**

| Method | CDVAE | FTCP | SLI2Cry |
|---|---|---|---|
| Match rate (%) | 45.43[23] | 69.89[23] | 84.66 |

Table 1 illustrates the reconstruction performance of SLI2Cry for the filtered MP-20 dataset (40,330 crystals): Under the strict criteria, the match rate of rescaled structures stands at 77.87%, suggesting that the combination of Eon's topology-based method and rescaling of the unit cell with modified GFN-FF effectively generates satisfactory initial guess structures. Following the optimization based on chemically meaningful geometry predicted with modified GFN-FF in step II, the match rate saw a 6.7% increase to 84.57%. Additionally, further structural refinement with the IAP during step III boosted the match rate to 92.55%. Direct refinement of the rescaled structures without step II only yielded an 86.83% match rate. This highlights the important role of the optimization targeting modified GFN-FF geometry in step (II) to the success of SLI2Cry. Given that SLICES maintains invariances by only encoding topology and composition, a match rate of 92.55% is impressive.

Under the loose setting, the match rates are as follows: rescaled structures: 91.36%, after $ZL^*$-optimization (step II): 94.05%, after IAP relaxation (step III): 94.95% (slight improvement). However, running IAP structural refinement directly on the rescaled structure (skip step II) caused a 0.78% decrease in match rate. This can be attributed to the relatively loose matching criteria in the loose setting, allowing for the inclusion of some problematic rescaled structures as matches. Subsequent IAP refinement of them yielded worse structures that then failed to meet the loose criteria.

Furthermore, we analyzed four representative cases where SLI2Cry faced challenges in reconstructing original crystal structures (Supplementary Note 2 and Supplementary Fig. 1). The findings indicate that further improving the accuracy and robustness of modified GFN-FF in step (II) could enhance the performance of SLI2Cry.

The reconstruction performances of previous methods were evaluated for the MP-20 dataset (45,229 crystals) under the loose setting[23], and their results were compared with SLI2Cry in Table 2. When applied to the 45,229 crystals within the MP-20 dataset, SLI2Cry achieved a match rate of 84.66%. This figure is lower than the 94.95% match rate observed on the filtered MP-20 dataset comprising 40,330 crystals. This decrease can be attributed to the inapplicability of SLI2Cry to 10.83% of the MP-20 dataset, primarily due to either high atomic numbers exceeding 86 or low dimensionality. Nevertheless, the achieved match rate of 84.66% still significantly surpasses that of CDVAE[23] (45.43%) and Fourier-transformed crystal properties (FTCP)[20] (69.89%). Note that FTCP lacks invariance to Euclidean transformations since it encodes absolute coordinates and lattice parameters. In contrast, SLI2Cry maintains invariances while achieving higher reconstruction performance than FTCP, primarily owing to SLI2Cry's strategy of combining an initial guess derived from graph theory with the optimization based on chemically meaningful geometry predicted by the modified GFN-FF. The reconstruction of the filtered MP-20 database (40,330 crystals) was completed within one hour on a workstation with 2 Xeon E5-2699v4 processors (2x22 cores, 2.2 GHz),

indicating SLICES is suitable to be integrated into inverse design pipelines of crystals.

Additionally, we evaluated the performance of SLI2Cry on the filtered MP-21-40, which comprises 23,560 materials with 21–40 atoms per unit cell from the Materials Project (see Methods section for details). Despite a minor performance decrease compared to that of the filtered MP-20 dataset (Table 1), SLI2Cry still accomplished high match rates of 87.88% (loose) and 83.73% (strict) on the filtered MP-21-40 (Supplementary Table 1). Notably, crystals with 1–40 atoms per unit cell account for 77.1% of all entries in Materials Project database, highlighting the broad applicability of SLI2Cry.

We also assessed SLI2Cry on 339 MOFs with 21–40 atoms per unit cell from the Quantum MOF database[38] (filtered QMOF-21-40). The match rates of 6.19% under loose criteria and 2.95% under strict criteria indicate the current limitation of SLI2Cry in reconstructing MOFs (Supplementary Note 3).

## Applying SLICES: Inverse design of direct narrow-gap semiconductors for optoelectronic applications

We showcase the application of SLICES for the inverse design of direct narrow-gap semiconductors targeting optoelectronic applications. The inverse design workflow consists of four stages (Fig. 3): (1) A general recurrent neural network (RNN)[39] was trained on the Materials Project[25] database to learn the syntax of SLICES strings; (2) A specialized RNN was then developed by tuning the general RNN using a dataset of direct narrow-gap semiconductors; (3) The specialized RNN was used to generate large volumes of SLICES strings, which were then reconstructed into crystal structures; (4) These crystal structures were screened to identify new direct narrow-gap semiconductors.

We set four design criteria: target bandgap, stability, composition novelty and structural uniqueness. Specifically, (1) direct bandgap (at Perdew-Burke-Ernzerhof (PBE)[40] level) $E_g^{PBE}$ = 0.325 (±0.225) eV, (2) energy above hull $E_{hull}$ < 50 meV/atom, (3) the composition must be different from entries in the Materials Project database, and (4) the structure should display low structural similarity to structures in training sets. We considered the tendency of PBE[40] to underestimate bandgap by an order of 0.6–1 eV for small bandgap crystals[41] when choosing the bandgap range. Moreover, direct bandgap enables applications of candidates in optoelectronics. Meanwhile, crystals with $E_{hull}$ < 50 meV/atom are assumed to be synthetically accessible[42] due to finite temperature effect and the error of density functional theory.

The task for material discovery using GMs is twofold: learning the syntax of the SLICES representation and learning topological/compositional features targeting key properties. We initially trained a general RNN on the "general dataset" (Fig. 3), which includes crystals structures from the Materials Project that satisfy four conditions: (1) 1–10 atoms in the unit cell, (2) formation energy $E_{form}$<0, (3) containing atoms with atomic number up to 86, and (4) without low-dimensional structural units. We fine-tuned a specialized RNN to learn features predictive of direct narrow bandgap. This was achieved by training it on crystal structures from the general dataset with a direct bandgap $E_g^{PBE}$ = 0.325 (±0.225) eV. Arús-Pous et al.[43] demonstrated that using randomized SMILES improves generative model performance over canonical SMILES. Therefore, we applied SLICES randomization (data augmentation) to both the general dataset (30,085 SLICES) and the transfer dataset (364 SLICES), resulting in 764,546 and 11,373 SLICES

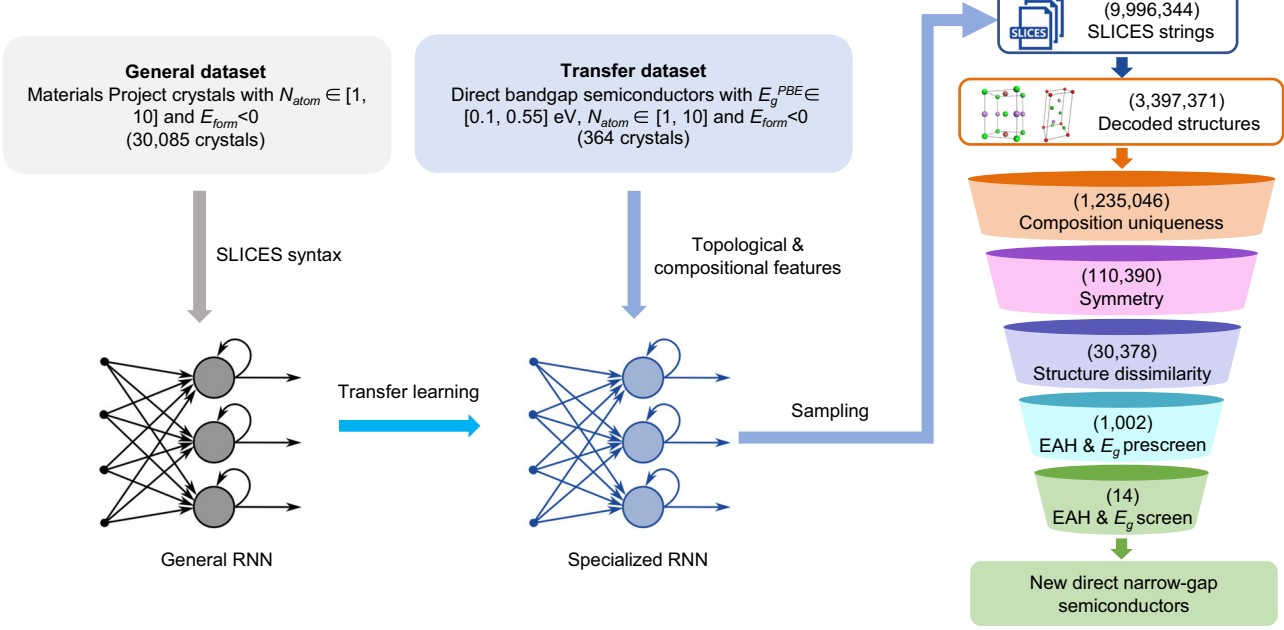

**Fig. 3 | The workflow for inverse design of direct narrow-gap semiconductors targeting optoelectronic applications.** The inverse design workflow started from training a general recurrent neural network (RNN) on the Materials Project database to learn the syntax of SLICES, followed by training a specialized RNN by tuning the general RNN using a dataset of direct narrow-gap semiconductors. Then, the specialized RNN was used to generate ~10 million SLICES strings, which were reconstructed into ~3.4 million crystal structures. These crystal structures were filtered to identify new direct narrow-gap semiconductors.

strings respectively. The randomization was achieved by arbitrary permutations of atom order and edge order in SLICES strings. The RNN architecture applied here was based on the work of Yuan et al.[6] (refer to ref. [6] for model details). Leveraging a one-hot encoding of SLICES, the general RNN and specialized RNN were trained for 10 epochs and 8 epochs, respectively (Supplementary Table 2).

We used the specialized RNN to generate ~10 million SLICES strings. Among them, ~3.4 million strings were decoded into crystal structures, while reconstruction was unsuccessful for ~6.6 million strings. This is primarily due to duplicated edges within these strings. This underscores the difficulties of RNNs in learning the complex syntax of long SLICES strings. State-of-the-art NLP architectures like Transformer[44] could help address this challenge, and is planned for future study. Through a multi-step high-throughput screening process, we identified 14 new direct narrow-gap semiconductors that met our design criteria. First, we removed candidates with compositions existing in the MP database, narrowing down to 1.24 million candidates. Next, to avoid duplicates from data augmentation, we kept the highest symmetry representatives of candidates with identical compositions, reducing to ~0.11 million candidates. We evaluated structural uniqueness between designed and training crystals using a dissimilarity value based on site coordination information[32]. Values near zero signify identical structures, whereas values surpassing 1 represent substantial structural differences. Subsequently, candidates with structural dissimilarity less than 0.75 (Materials Project's threshold) to training structures were eliminated, leaving ~0.03 million candidates. It's worth noting that about 27.5% candidates have a dissimilarity value above 0.75, indicating the SLICES-based RNN model can design crystals that are unlikely to be discovered by elemental substitution. Furthermore, we removed candidates with M3GNet IAP-predicted energy above hull $E_{hull}^{IAP} \geq$ 50 meV/atom. Then, using Atomistic Line Graph Neural Network (ALIGNN)[45] model jv_optb88vdw_bandgap_alignn[46], we eliminated candidates with $E_g^{ALIGNN} <$ 0.1 eV (less likely to be a semiconductor), narrowing down the search space to 1,002 candidates. Finally, after performing structural relaxation (MPRelaxSet[33]) and band structure calculation at PBE[40] level

using Vienna Ab-initio Simulation Package (VASP)[47,48], we discovered 14 new direct narrow-gap semiconductors meeting our design criteria (structures, bandgaps and $E_{hull}$ are shown in Fig. 4). The band structures of these materials are depicted in Fig. 5.

A workstation with dual Xeon E5-2699v4 CPU (2x22 cores, 2.2 GHz) and a NVIDIA RTX 2080 Ti GPU was employed to run the inverse design scheme (Supplementary Note 4). In total, 14 potentially synthetically accessible direct narrow-gap semiconductors with unique compositions and structures were inversely designed in less than 11 days on this workstation.

## Benchmarks on material generation and property optimization

To compare the generation and property optimization performance of SLICES-based inverse design frameworks with FTCP[20] and CDVAE[23], we trained an unconditional RNN (termed as ucRNN) and a conditional RNN (denoted as cRNN) on the filtered MP-20 dataset (see Methods section and Supplementary Table 2 for details).

We evaluated the material generation performance of the SLICES-based ucRNN model using structural and compositional validity metrics proposed by Xie et al.[23] Specifically, a structure is deemed valid if the minimal atomic distance exceeds 0.5 Å, while compositional validity requires overall charge neutrality as determined by Semi-conducting Materials from Analogy and Chemical Theory[49] v2.5.2. We sampled 10,000 SLICES strings using the ucRNN model and evaluated the validity metrics on 9,428 reconstructed crystals (see Methods section for details). Our method achieves a higher validity than FTCP, while achieving a similar validity as CDVAE (Table 3).

We evaluated the property optimization performance of the SLICES-based cRNN model using the success rate proposed by Xie et al.[23] Specifically, the success rate (SR) is defined as the percentage of crystals achieving 5, 10, and 15 percentiles of the formation energy distribution of the training set. The goal of property optimization is to minimize the formation energy per atom for the generated materials. We sampled 1000 SLICES strings using the cRNN model and evaluated the SR on 782 reconstructed crystals (see Methods section for details). Our method considerably outperforms CDVAE and FTCP (Table 3),

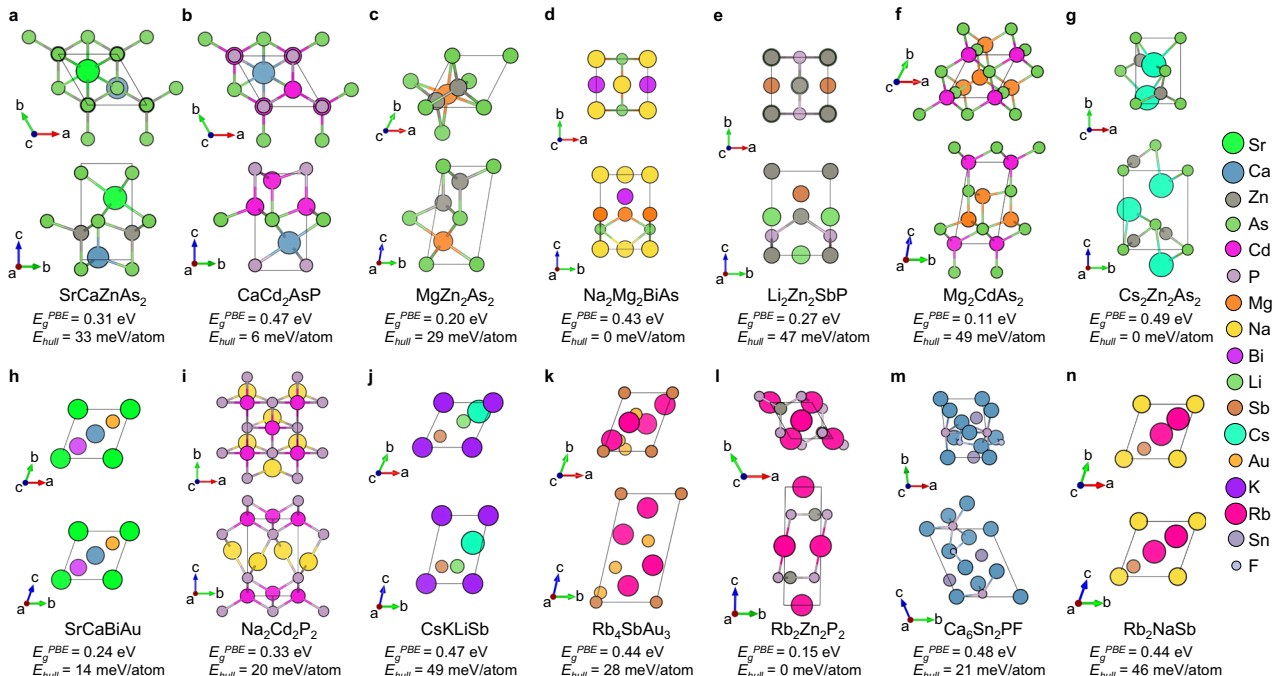

**Fig. 4 | Top and side views of 14 new direct narrow-gap semiconductors.** For each material, the values of bandgap at Perdew-Burke-Ernzerhof (PBE) level ($E_g^{PBE}$) and energy above hull ($E_{hull}$) are provided.

showcasing the potential of SLICES for inverse design of solid-state materials.

## Discussion

We present SLICES, a string-based, invertible and invariant crystal representation. Analogous to SMILES representation for molecules, SLICES encodes the topology and composition of crystal structures into strings. SLI2Cry reconstructs input crystal structures in three steps: (1) Initial structure generation with graph theory techniques; (2) Optimization using chemically meaningful geometry predicted with modified GFN-FF; (3) Structural refinement with the M3GNet IAP. SLI2Cry outperforms past methods in reconstructing input structures while still preserving invariances. To our knowledge, SLICES is the first invertible crystal representation that satisfies full invariances. Utilizing SLICES-based RNN models, we inversely designed new direct narrow-gap semiconductors with unique compositions and structures. Moreover, SLICES-based inverse design framework considerably outperforms past approaches in generating materials with a desired property.

While SLICES' current reconstruction scheme is the best achievable now, further improvements in reconstruction performance are possible. First, graph theory techniques utilized in step (I) are inadequate to handle low-dimensional crystals like molecular or layered crystals. A hierarchical graph approach that treats low-dimensional structural units as nodes in quotient graphs could potentially address this, which requires future work. Second, the modified GFN-FF in step (II) could be replaced with a universal force field that covers the entire periodic table and provides high-accuracy bond length/angle predictions from geometry-independent inputs.

While SLICES can encode the chemical connectivity of MOFs, SLI2Cry faces challenges for reconstructing MOFs from SLICES strings. To develop an invertible representation for MOFs (termed MOF-SLICES), we propose encoding structural building units (SBUs) like organic ligands and metal clusters as single nodes when constructing quotient graphs. The SBU symbols can be represented by their indices in a predefined SBU database. For rebuilding MOFs from MOFSLICES strings, we can build upon the topology-based MOF construction algorithm proposed by Boyd and Woo[37]. This hierarchical graph

approach that simplifies SBUs into quotient graph nodes could potentially enable MOF reconstruction, which is planned for future work.

Utilizing other pre-trained ALIGNN models for prescreening and JARVIS-Tools[50] for ab initio validation, this inverse design framework can be adapted for discovering various functional materials, such as topological materials and high-$T_C$ superconductors. Just as SMILES enables advanced NLP models like Transformer[44] for inverse molecular design, as demonstrated by Bagal et al.[51] with MOLGPT, a GPT[52]-style decoder for de novo molecular discovery. Analogously, SLICES could empower GPT decoders to inversely design solid-state materials by representing crystals as strings. In summary, as a string-based, invertible, and invariant crystal representation, SLICES showcases potential as a useful tool for the inverse design of functional crystalline materials.

## Methods

### Eon's topology-based method

**Graph theoretical terms.** Here, we first introduce some basic graph theoretical terms that will be mentioned frequently below.

**Cycles** in a graph are non-empty paths in which only the starting and ending nodes are the same.

**Co-cycles** are sets of edges, if removed, would cut a connected graph into two disjoint subgraphs.

**Constructing edges vectors.** For an labeled quotient graph with $m$ edges *and* $n$ nodes, Delgado-Friedrichs & O'Keeffe[36] proposed that edges vectors in fractional coordinates of the unit cell, $\Omega$, is obtained as follows:

$$\Omega = B^{-1} \cdot \alpha \qquad (4)$$

where $B$ is a $m \times m$ matrix of cycle/co-cycle basis vectors and $\alpha$ is a $m \times 3$ matrix of lattice/co-lattice vectors.

The cycle basis of a graph represents an irreducible depiction of all possible cycles within it. A cycle basis vector is a sum of the edges (accounting for orientation) within it. For instance, $e_1$ and $-e_2$ form a

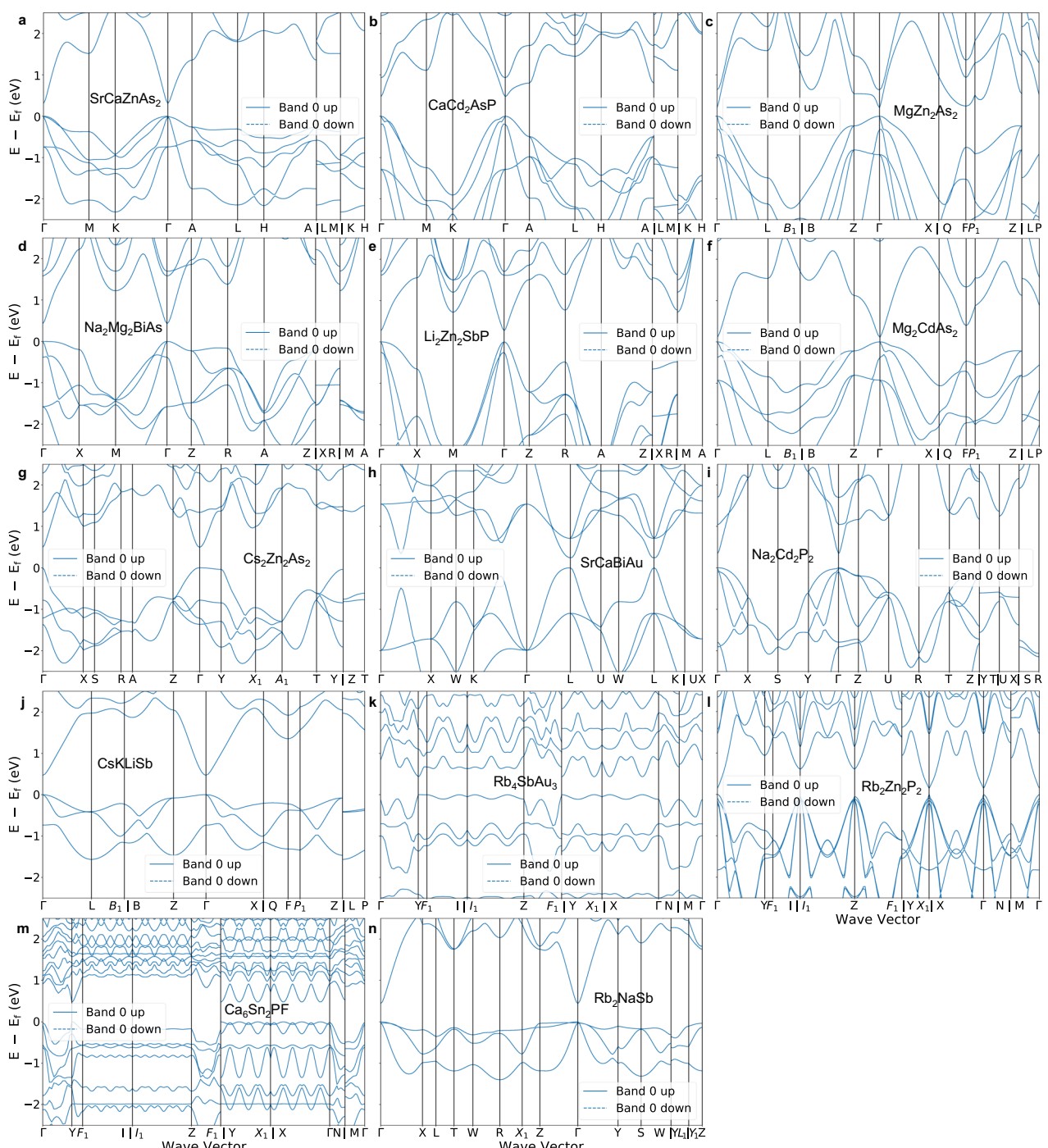

**Fig. 5 | Perdew-Burke-Ernzerhof (PBE) band structures of 14 promising direct narrow-gap semiconductor candidates.** Source data are provided as a Source Data file.

cycle (Fig. 1b) which is represented in terms of edges $(e_1, e_2, e_3, e_4)$ as $(1, -1, 0, 0)$. The minimum spanning tree algorithm was used to construct the cycle basis for labeled quotient graphs of crystals. That is, the cycle basis of a graph can be systematically constructed by selecting cycles formed by combination of a path in the minimum spanning tree and an edge outside the tree. For connected graphs, the basis of the cycle space consists of $m - n + 1$ vectors, which correspond to the first $m - n + 1$ rows of matrix $B$. The remaining $n - 1$ rows of matrix $B$ consist of co-cycle basis vectors. The co-cycle basis can be constructed by summing all outward oriented edges (flip the edge if it is inward oriented) from the first $n - 1$ nodes of the graph. For

instance, a co-cycle basis vector for the graph in Fig. 1b is the sum of all outward oriented edges of node $C_0$, i.e., $(1, 1, 1, 1)$.

Equation (4) shows that $B$ transforms the edge vectors in $\Omega$ to their lattice/co-lattice representation in $\alpha$. The first $m - n + 1$ rows of matrix $\alpha$ are called "lattice vectors" $L$ and the remaining $n - 1$ rows correspond to co-lattice vectors $L^*$. A lattice vector translates points within the unit cell to an equivalent point in a repeating unit of the crystal lattice. The lattice vector of a cycle can be constructed by summing edge labels (translation vectors) in a cycle. For instance, in Fig.1b, $e_1$ (translation vector: $(0,0,0)$) and $-e_2$ (translation vector: $(-1,0,0)$) form a cycle, so the lattice vector of this cycle is $(-1,0,0)$.

**Table 3 | Generation performance and property optimization performance**

| Method | Generation performance (%) | | Property optimization performance (%) | | |
|---|---|---|---|---|---|
| | Structural validity | Compositional validity | SR5 | SR10 | SR15 |
| FTCP[20,23] | 1.55 | 48.37 | 2.00 | 4.00 | 5.00 |
| CDVAE[23] | 100.0 | 86.70 | 78.0 | 86.0 | 90.0 |
| SLICES | 99.72 | 84.43 | 97.4 | 99.2 | 99.6 |

A co-lattice vector is analogous to a lattice vector, except defined on a co-cycle rather than a cycle. Thus, it can be constructed by summing the edges vectors in a co-cycle. The co-lattice vectors $L^*$ in any barycentric embedding is a zero matrix. This is because all nodes are the center of mass of their neighboring nodes, resulting in the sum of the edges vectors of the first $n-1$ nodes being null vectors. To this end, co-lattice vectors of an embedding can be used to quantify the deviation of the embedding from its barycentric embedding. By utilizing co-lattice vectors $L^*$ as variables in an objection function, we can optimize the fractional coordinates of nodes to match the geometry predicted by modified GFN-FF.

Having constructed $B$ and $\alpha$, then edge vectors $\Omega$ can be obtained using Eq. (4). Once $\Omega$ has been determined, the fractional coordinates of all nodes can then be readily calculated.

In the case of labeled quotient graph of diamond in Fig. 1b, we can construct its $B_{dia}$ and $\alpha_{dia}$ as follow:

$$B_{dia} = \begin{pmatrix} 1 & -1 & 0 & 0 \\ 1 & 0 & -1 & 0 \\ 1 & 0 & 0 & -1 \\ 1 & 1 & 1 & 1 \end{pmatrix}, \alpha_{dia} = \begin{pmatrix} -1 & 0 & 0 \\ 0 & -1 & 0 \\ 0 & 0 & -1 \\ 0 & 0 & 0 \end{pmatrix} \quad (5)$$

Using Eq. (4), we can obtain $\Omega(0)_{dia}$ as:

$$\Omega(0)_{dia} = \begin{pmatrix} \frac{1}{4} & \frac{1}{4} & \frac{1}{4} & \frac{1}{4} \\ -\frac{3}{4} & \frac{1}{4} & \frac{1}{4} & \frac{1}{4} \\ \frac{1}{4} & -\frac{3}{4} & \frac{1}{4} & \frac{1}{4} \\ \frac{1}{4} & \frac{1}{4} & -\frac{3}{4} & \frac{1}{4} \end{pmatrix} \begin{pmatrix} -1 & 0 & 0 \\ 0 & -1 & 0 \\ 0 & 0 & -1 \\ 0 & 0 & 0 \end{pmatrix} = \begin{pmatrix} -\frac{1}{4} & -\frac{1}{4} & -\frac{1}{4} \\ \frac{3}{4} & -\frac{1}{4} & -\frac{1}{4} \\ -\frac{1}{4} & \frac{3}{4} & -\frac{1}{4} \\ -\frac{1}{4} & -\frac{1}{4} & \frac{3}{4} \end{pmatrix} \quad (6)$$

To calculate fractional coordinates of nodes, we can first place node $C_0$ at (0,0,0). The fractional coordinate of node $C_1$ can then be obtained by adding the first row in $\Omega(0)_{dia}$ to the position of $C_0$, resulting in $(-1/4, -1/4, -1/4)$.

**Constructing the metric tensor.** To construct an embedding of labeled quotient graph in Cartesian space, we have to calculate metric tensor $Z = \begin{pmatrix} \mathbf{a} \cdot \mathbf{a} & \mathbf{a} \cdot \mathbf{b} & \mathbf{a} \cdot \mathbf{c} \\ \mathbf{b} \cdot \mathbf{a} & \mathbf{b} \cdot \mathbf{b} & \mathbf{b} \cdot \mathbf{c} \\ \mathbf{c} \cdot \mathbf{a} & \mathbf{c} \cdot \mathbf{b} & \mathbf{c} \cdot \mathbf{c} \end{pmatrix}$. Eon[26] showed that the metric tensor of the barycentric embedding corresponding to a labeled quotient graph can be obtained as follow:

$$Z = L_A \cdot P \cdot (L_A)^t \quad (7)$$

where $L_A$ is a $3 \times m$ matrix containing cycle basis vectors, and $P$ is an orthogonal projection matrix. The kernel $K$ of projection matrix $P$ is a basis for cycles whose edge labels sum to the null vector. The use of this projection matrix ensures that when constructing the metric tensor from the edge space, cycles with translation vectors summing to zero are mapped to the zero vector. $P$ is obtained as follow[53]:

$$P = I - K^t \cdot (K \cdot K^t)^{-1} \cdot K \quad (8)$$

If the cycle basis of a labeled quotient graph has a rank of 3, the projection matrix reduces to the identity matrix, $I$. For instance, the cycle space of the labeled quotient graph in Fig. 1b has 3 basis vectors. For the diamond structure in Fig. 1b, we can obtain its metric tensor $Z_{dia}$ using Eq. (7) as:

$$Z_{dia} = \begin{pmatrix} 1 & -1 & 0 & 0 \\ 1 & 0 & -1 & 0 \\ 1 & 0 & 0 & -1 \end{pmatrix} \cdot I \cdot \begin{pmatrix} 1 & 1 & 1 \\ -1 & 0 & 0 \\ 0 & -1 & 0 \\ 0 & 0 & -1 \end{pmatrix} = \begin{pmatrix} 2 & 1 & 1 \\ 1 & 2 & 1 \\ 1 & 1 & 2 \end{pmatrix} \quad (9)$$

As a result, the lattice parameters of its barycentric embedding are $a = b = c = \sqrt{2}, \alpha = \beta = \gamma = 60°$. These parameters properly describe the primitive cell of diamond, except the length of the lattice vectors needs to be rescaled, owing to the barycentric embedding is obtained in a pure mathematical way.

**Constructing barycentric/non-barycentric embeddings.** Once the metric tensor $Z$ and edge vectors $\Omega(L^*)$ have been determined, the barycentric/non-barycentric Cartesian embeddings of a labeled quotient graph can be easily calculated.

## Modification of GFN-FF

GFN-FF is implemented in the semiempirical extended tight-binding (XTB)[54] package. We built a new module, xtb_io_reader_top, to accept neighbor lists as input and store the neighbor list in topo%nb. Moreover, we modified xtb_gfnff_ini, xtb_gfnff_ini2, xtb_gfnff_rab, xtb_gfnff_setup to initialize newGFFCalculator with topo%nb (without relying on Cartesian coordinates of atoms (mol%xyz)). Additionally, instead of calculating the coordination number of atoms (cn(i)) with Cartesian coordinates of atoms, cn(i) is set to the normal coordination number of that atom type (param%normcn(mol%at(i))). All modifications of the XTB package are shown in the forked XTB repository (https://github.com/xiaohang007/xtb).

Since GFN-FF implemented in XTB is designed for finite systems such as molecules or clusters, we simulate periodic boundary conditions using finite clusters ($3 \times 3 \times 3$ supercells) extracted from crystals. Specifically, the modified GFN-FF takes the neighbor list of the finite cluster ($3 \times 3 \times 3$ supercell) constructed based on the labeled quotient graph as input, and outputs equilibrium bond lengths/angles as well as relevant force constants of the finite cluster. Then, the equilibrium bond lengths/angles and relevant force constants corresponding to the central unit cell of $3 \times 3 \times 3$ supercell cluster are extracted and used to optimize the non-barycentric embedding in step (II) of SLI2-Cry (Fig. 2).

## Filtered MP-21-40 dataset

MP-21-40 comprises 24,959 materials with 21-40 atoms per unit cell from the Materials Project database. In MP-21-40, we select materials with formation energy smaller than 2 eV/atom and energy above the hull smaller than 0.08 eV/atom to exclude unstable materials, following Xie et al.[23]. After excluding crystals containing atoms with atomic numbers beyond 86 and those with low-dimensional structural units, the filtered MP-21-40 dataset consists of 23,560 crystals.

## ucRNN/cRNN Models for SLICES String Generation

The ucRNN model was trained on the filtered MP-20 dataset (40,330 SLICES). We applied data augmentation to the filtered MP-20 dataset, resulting in 2,009,115 SLICES strings. The RNN architecture applied here is the same with the RNN models used in the inverse design of direct narrow-gap semiconductors (Supplementary Table 2). The ucRNN was trained for 10 epochs. We sampled 10,000 SLICES strings using the ucRNN model. However, the majority of these SLICES strings contained duplicated edges that impeded reconstruction by SLI2Cry,

owing to the difficulties of RNNs in learning the complex syntax of long SLICES strings. Using advanced NLP architectures like Transformer[44] could help address this challenge and is planned for future work. A simple workaround applied in this study was removing all duplicated edges to correct syntax errors, enabling successful reconstruction of 9428 materials from the 10,000 sampled strings. We then evaluated the validity metrics on these 9428 generated structures to assess the ucRNN's performance.

The cRNN model was also trained on the filtered MP-20 dataset for controlled generation of crystals with desired formation energy. The model schematic of cRNN for training and generation is given in Supplementary Fig. 2a. For training, formation energies of crystals in MP-20 were passed as conditions alongside the SLICES string. The architecture of the cRNN model is illustrated in Supplementary Fig. 2b. To enable conditional generation, we extended the ucRNN with an additional dense layer that transforms the user-specified formation energy into a tensor. The concatenation of this tensor with the embedding tensor of SLICES is fed into a 3-layer stacked gated recurrent unit (GRU). The cRNN was also trained for 10 epochs (Supplementary Table 2).

For generation, we input a desired formation energy to the model to sample crystals. To generate crystals with minimal formation energy, we sampled 1000 SLICES strings for each of the formation energy targets (−3.0, −4.0, −4.5, −5.0, and −6.0 eV/atom). After removing duplicated edges in sampled strings, we used SLI2-Cry to reconstruct the corresponding crystals. The distribution of formation energy (predicted by M3GNet) of reconstructed crystals under these targets are depicted in Supplementary Fig. 2c. As seen in Supplementary Fig. 2c, the distribution of formation energy with target = −3.0, −4.0, −4.5 eV/atom is generally centered around the desired value, when taking into account the deviations between M3GNet predictions and PBE calculations. However, setting the target to lower values (−5.0, −6.0 eV) had an adverse impact, owing to the scarcity of training data samples exhibiting formation energies below −4.5 eV/atom (Supplementary Fig. 2c). In summary, the lowest mean formation energy predicted by M3GNet was achieved using a target of −4.5 eV/atom. Based on this observation, formation energies (at PBE level) of crystals generated with a target of −4.5 eV/atom were used to evaluate the success rate of property optimization.

## Software implementation

SLICES has been implemented as a Python package. It is published under the GNU Lesser General Public License v2.1, which is an open-source license that is recognized by the Open Source Initiative. The operating system needed for compilation and execution is GNU/Linux. For ease of installation and reproduction of the reconstruction benchmark and the inverse design case study, a Docker image with pre-installed SLICES v1.4 package, modified XTB (commit: 0fcba9e)[54] package, M3GNet v.0.2.4, Pymatgen[33] v.2022.11.7, Pytorch v.1.13.0, ALIGNN[45] v.2023.1.10, Jarvis-tools[50] v2023.1.8 is provided. Note that VASP[47,48] requires a commercial license and is not distributed in this Docker image.

## Reporting summary

Further information on research design is available in the Nature Portfolio Reporting Summary linked to this article.

## Data availability

The inverse design data of direct narrow-gap semiconductors and the data for reconstruction, material generation, and property optimization benchmarks are available at Figshare[55] (https://doi.org/10.6084/m9.figshare.22707472, Version 2). Source data are provided with this paper.

## Code availability

The SLICES source code is available on GitHub (https://github.com/xiaohang007/SLICES). The SLICES documentation is hosted at https://xiaohang007.github.io/SLICES/. SLICES v1.4[56] (https://doi.org/10.5281/zenodo.8421021) was used to generate all results in this work. A Docker image containing pre-installed SLICES and dependencies is available on Docker Hub (docker pull xiaohang07/slices:v3) and Figshare[57] (https://doi.org/10.6084/m9.figshare.22707946, Version 1) to facilitate reproducibility. The modified XTB package (commit: 0fcba9e) can be found at https://github.com/xiaohang007/xtb.

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

## Acknowledgements

L.W. acknowledges National Key Projects for Research and Development of China (Grant No. 2022YFA1204700 and 2021YFA1400400), the National Natural Science Foundation of China (Grant No. 12074173), the Program for Innovative Talents and Entrepreneur in Jiangsu (Grant No. JSSCTD202101) and Natural Science Foundation of Jiangsu Province (Grant No. BK20220066). H.X. acknowledges the support from the 6th Young Elite Scientist Sponsorship Program by China Association for Science and Technology (Grant No. 2020QNRC001), National Natural Science Foundation of China (Grant No. 22203066). L.Z. acknowledges the support from the National Natural Science Foundation of China (Grant No. 12002271). Many useful discussions with Prof. Hisashi Naito from Nagoya University are acknowledged.

## Author contributions

H.X. conceived the idea. L.Z., L.W., X.C. and Y.C. designed the work. R.L. implemented the models. Y.C. and X.S. performed the analysis. H.X. and R.L. wrote the manuscript. L.W., L.Z., X.C. and Y.C. contributed to the discussion and revision.

## Competing interests

The authors declare no competing interests.
