## [Peer Review File · Nature Communications]

An invertible, invariant crystal representation for inverse design of solid-state materials using generative deep learningREVIEWER COMMENTS

Reviewer #1 (Remarks to the Author):

The manuscript describes the features of a new cheminformatics format, so called SLICES, designed to provide an inline representation of non-discrete chemical species (i.e., polymers, non-molecular crystals ...), thus extending the capabilities of other available, widely spread used formats such as INchi or SMILES, which present the important limitation of being restricted to molecular entities of finite size.

I think that this new format could become a potentially powerful tool for the representation of such non-discrete species, finding applications in fields such as solid state chemistry, organic polymers or metal organic frameworks and hence of interest for a broad audience. The definition and wide-spread use of a format able to faithfully represent the chemical connectivity in 1D, 2D and 3D extended chemical systems without the need of including crystallographic data (unit cell, symmetry, coordinates ...) will probably help a lot in the chemical description and informatics treatment of such compounds and I think the proposal presented in this work is a very good candidate to become such format: its definition is quite simple and clear and, from my point of view, its integration in different cheminformatics tools should not be too difficult.

The manuscript also describes a procedure to regenerate the 3-D crystal structure of a compound starting only with the chemical connectivity indicated in a SLICES string, which comprises three steps: the generation of the polymeric chemical graph, the creation of a chemically reasonable 3-D model using theoretical calculations (GFN-FF) and further optimization of such model with a deep learning method. In fact, the explanation of such procedure makes the main body of the manuscript. The authors claim a success rate of almost 95% for matching the theoretically regenerated structure with that initially used to derive the SLICES string: this is what the authors mean by using the word "invertible", as defined in the introduction. I am not sure if this is the best word to express the concept, but it is OK if it is generally used with this meaning in the crystal engineering research field.

Another section of the manuscript is devoted to the generation of a very large number of possible chemical connectivity schemes, represented as SLICES strings, trying to build theoretical crystal structures from them and, if that step succeeds, analysing their predicted properties, in particular the band gap to identify potential semiconductors. The analysis ends with a bunch of candidates. This is an example of what the SLICES format may be used for.

As told above, most of the paper is devoted to the procedure for the regeneration of the crystal structure from SLICES with just a short section devoted to the description of the format itself. This is understandable since the main interest of the authors seems to be the application of the format to crystal design of materials ("in silico design") but I think SLICES format could be rather useful even just for descriptive applications, I am specially thinking of it as a way to store the chemical connectivity in MOF databases. Because of it, I think that the very short section devoted to the description of the format (lines 89-97) should be extended, stressing its features. The need for specifying the whole set of atoms in the unit cell avoids the inclusion of symmetry codes thus simplifying the rules for building the string at the cost of getting a representation that is not very compact (for example a cubic MOF with, say $Z = 48$, even with very few atoms in the asymmetric unit would generate a SLICES with several hundred atoms and a even higher number of edges). This implies repetition of symmetry-related atoms and bonds but redundant information is for sure better than lack of information and nowadays simplicity and reliability of the result are much more important than the length of the string.

It is not clearly told if the software briefly mentioned in "software implementation" (lines 443-447) and in "code availability" (lines 454-459) is for generating the SLICES from experimental crystal structures, for generating theoretical structures from SLICES or both. Presumably, it is for both and, in such case, the pieces of software executing each of the two functions (crystal -> SLICES and SLICES -> crystal) should be clearly differentiated. For example, for descriptive purposes, only

the first part would be necessary. This first part is, with no doubt, much easier than the second but perhaps the term "straightforward" (used in line 99) is not fully appropriate and I think that the crystal -> SLICES task deserves some extra detail. I guess that this portion of the software is able to generate the SLICES from crystallographic information (CIF file???) but this is not explicitly told. Also, how is the chemical connectivity generated, just taking the `_geom_bond_*` items provided by the CIF? Calculating all distances between pairs of atoms and deciding which among them represent chemical bonds by defining some threshold? The authors just mention "EcoN" and "pymatgen" with no further details.

It is important to say the license under which the software is available (GPL???). Also it should be stated the operating system needed for compilation and execution (GNU/Linux, MS Windows, MAC, any of them?).

There is a lot of confusion in the paper between the format and the reconstruction routine, sometimes "SLICES" is used to refer to the format, in other places "SLICES" is used to refer to the rebuilding procedure. Both are not the same thing and I think the acronym "SLICES" should be used exclusively to design just the format and may be some other acronym (I will use "SLItoCry" as example from here onwards) to refer to the rebuilding procedure. The authors should set it more clear when they are talking about one thing or about the other.

For example, line 234: "SLICES was unable to be applied to 10.83% due to ...". This is not true. SLICES (the format) can be applied without problems to all structures, what cannot be applied is SLItoCry.

Also in line 324: "SLICES representation covers the majority of elements ..." is not correct, the representation cover ALL elements of the periodic table. Again, it is SLItoCry or more precisely GNFF, what does not present that full coverage.

Line 26 and line 326: "SLICES reconstructs ...". The format on its own does not reconstruct anything: it is just used as the starting point for the reconstruction.

Line 66 and line 329: "SLICES outperforms past methods ...". SLICES is a format, not a method, the method is SLItoCry.

The success rate of SLItoCry in the chosen benchmark (MP-20) is, in fact, impressively high but it must be stressed that the 20 atoms per unit cell limitation is quite restrictive and surely leaves out a lot of structures that would be interesting to be tested, more notably those including organic portions (COFs, MOFs) that are likely to be underrepresented specially if they are highly symmetric, which implies an important bias to the nature of the analysed set. It would be interesting to see how SLItoCry works with this kind of compounds maybe testing a set of (say) a few hundreds COFs/MOFs with more than 20 atoms in the unit cell.

A minor question is the appearance of many acronyms that have not been defined (CDVAE, GNNFF, FTCP, RNN, ...). They should be defined in parenthesis the first time they appear. Even SMILES and SLICES itself are defined in the abstract but not in the body of the paper (the definition should be repeated in their first appearance in the introduction).

Reviewer #2 (Remarks to the Author):

the authors developed a framework to inverse design crystals using an invertible crystallographic representation and 3-step structural optimization methods. The representation is a string-based crystallographic representation that satisfies both invertibility and symmetry invariances. The authors showcase the application of this framework to direct narrow-gap semiconductors. This topic is of interest to the material informatics field and the framework showed improvement over past studies. However, I have concerns regarding the metrics used to validate the results.

1. In Table 1 and Table 2, the match rate is used as a tool to demonstrate the effectiveness of the 3-step structural optimization process and benchmark the framework with other two studies. It is

not clear how the match rate is calculated. How is it defined? How many data points are used as the training data, testing data, and validation data?

2. Matching rate is a good metric in comparing the reconstruction performance of the generative algorithms. However, aside from the reconstruction performance, generation performance, and property optimization performance are also important metrics for crystal inverse design algorithms. In the FTCP and CDVAE study, the validity rate and success rate are also reported to show the generation performance of the models. It will be interesting to see SLICE's comparison with the other two studies.

Minor point: In Figure 3, it shows that the generated crystals (sampled from the latent space) passed through a series of filters to be down-selected as the candidates. Will the addition of a property prediction branch to your RNN to shape the latent space make this step more efficient?

Reviewer #3 (Remarks to the Author):

In their manuscript "An invertible, invariant crystallographic representation for inverse design of solid-state materials using generative deep learning" Xiao et al. present a string representation method to describe solid state crystal structures. The authors aim to develop a string representation as successful as SMILES while overcoming its shortcomings, most importantly, inability to represent covalent networks intrinsic to solid state crystal structures. Quotient graphs have been used to analyze such structures before (see for example Gao et al., 2020, doi:10.1038/s41524-020-00409-0), but for me the most interesting part of the manuscript is the employment of a mechanism to invert the representation by reconstructing crystal structures. The authors demonstrate an impressive fidelity of such reconstruction, as well as illustrate the usability of their representation for the design of novel materials.

I have the following comments, questions and suggestions about the manuscript:

1. Why the authors have chosen a string representation with one-hot encoding as input to deep learning? When underlying data are graphs, using them directly as inputs in graph neural networks seems more natural to me. I believe the manuscript could benefit from an explanation of benefits of such choice.

2. The authors demonstrate a high success rate for structure reconstruction from SLICES. However, it would also be interesting to see the analysis of failures, even if just a couple of them.

3. Coming from crystallographic background I find the usage of some terms confusing. First of all, when seeing "symmetry" (for example, line 23) I tend to think about crystal symmetry, but it seems that this term is used in other sense in most of the text, except probably in line 295. I would suggest explaining the meaning of "symmetry" in more detail. Then in line 324 the authors use term "crystallographic representation" where I think "crystal representation" is more appropriate.

4. Some parts of the text present claims that are not very well based, I would suggest rephrasing them, or removing them altogether. In the abstract (line 31) and introduction (line 70) the authors claim that SLICES has the potential to "become a standard tool", I think it is too early to make such a claim. In line 242 the authors talk about computational efficiency of the reconstruction scheme. In my opinion, a scheme which requires crystal structure reconstruction with forcefields is quite computationally expensive. I believe such claim is appropriate only when comparing reconstruction times with other representations. Also I would suggest rephrasing line 321 to avoid using word "democratize" which is very unclear in this context. Please as well remove words "user-friendly" from line 443, as such claim is inappropriate in primary sources.

5. I applaud the authors' choice to upload the used software and datasets to FigShare, but I

suggest improving provenance and reproducibility of your research. Versions for all pieces of software and datasets have to be indicated. Please cite Git tag or commits for SLICES and the modified XTB package. FigShare uploads also have versions, please cite them as well, because future uploads may cause ambiguity.

6. Certain parts of the results section could benefit from more details. It should be explained what term "augmented" in line 281 means. In lines 290-291 it should be explained why such a decrease happened. When talking about dissimilarity measure in line 299 it would be nice to explain what do lower and higher values mean. Figure 3 could include dataset sizes.

7. Some minor points:

- * It is uncommon to start sentences with "And ...", I suggest avoiding such constructions.
- * Abbreviation "RNN" (line 252 for example) is not explained anywhere in the text.
- * "InChI" is written incorrectly in line 37.
- * Are the URLs in lines 450 and 452 meant to be identical?
- * Please cite git commit in reference 24.
- * Please elaborate references 37 and 44, at least authors and URLs are needed.
- * In Table 2, why is the match rate of SLICES different from the one provided in Table 1?
- * "Euclidian" in line 102 should be spelled as "Euclidean".
- * "Systematically" in line 111 should be spelled as "systematic".
- * Generally I find it difficult to understand where figure captions end and the regular text begins.
- * Chemical formulas are not necessary in figure captions of Figures 4 and 5.

Point-by-point response to the reviewers' comments

Reviewer #1

The manuscript describes the features of a new cheminformatics format, so called SLICES, designed to provide an inline representation of non-discrete chemical species (i.e., polymers, non-molecular crystals ...), thus extending the capabilities of other available, widely spread used formats such as INchi or SMILES, which present the important limitation of being restricted to molecular entities of finite size.

I think that this new format could become a potentially powerful tool for the representation of such non-discrete species, finding applications in fields such as solid state chemistry, organic polymers or metal organic frameworks and hence of interest for a broad audience. The definition and wide-spread use of a format able to faithfully represent the chemical connectivity in 1D, 2D and 3D extended chemical systems without the need of including crystallographic data (unit cell, symmetry, coordinates ...) will probably help a lot in the chemical description and informatics treatment of such compounds and I think the proposal presented in this work is a very good candidate to become such format: its definition is quite simple and clear and, from my point of view, its integration in different cheminformatics tools should not be too difficult.

The manuscript also describes a procedure to regenerate the 3-D crystal structure of a compound starting only with the chemical connectivity indicated in a SLICES string, which comprises three steps: the generation of the polymeric chemical graph, the creation of a chemically reasonable 3-D model using theoretical calculations (GFN-FF) and further optimization of such model with a deep learning method. In fact, the explanation of such procedure makes the main body of the manuscript.

Response: Thank you very much for your positive remarks and kind suggestions. We improved our work accordingly. Please find below our point-to-point responses (in blue) to your comments (in black). The revisions are shown in blue color in the revised manuscript.

1. The authors claim a success rate of almost 95% for matching the theoretically regenerated structure with that initially used to derive the SLICES string: this is what the authors mean by using the word "invertible", as defined in the introduction. I am not sure if this is the best word to express the concept, but it is OK if it is generally used with this meaning in the crystal engineering research field.

Response: Thank you very much for your insightful comments. “Invertible” and “invertibility” are generally used in crystal engineering. For example, (1) “Invertible Image-Based 3D Representations for Crystal Structures” was used to describe the image-based crystal representation proposed by Noh *et al.* (*Matter* 1.5 (2019): 1370-1384); (2) In a minireview titled “Machine-enabled inverse design of inorganic solid materials: promises and challenges” (*Chemical Science* 11.19 (2020): 4871-4881), it was mentioned that “The first two issues (invertibility and invariance) correspond to the characteristics of representations, while the third issue, chemical diversity, is related to the training data”.

2. Another section of the manuscript is devoted to the generation of a very large number of possible chemical connectivity schemes, represented as SLICES strings, trying to build theoretical crystal structures from them and, if that step succeeds, analysing their predicted properties, in particular the band gap to identify potential semiconductors. The analysis ends with a bunch of candidates. This is an example of what the SLICES format may be used for.

As told above, most of the paper is devoted to the procedure for the regeneration of the crystal structure from SLICES with just a short section devoted to the description of the format itself. This is understandable since the main interest of the authors seems to be the application of the format to crystal design of materials ("in silico design") but I think SLICES format could be rather useful even just for descriptive applications, I am specially thinking of it as a way to store the chemical connectivity in MOF databases. Because of it, I think that the very short section devoted to the description of the format (lines 89-97) should be extended, stressing its features. The need for specifying the whole set of atoms in the unit cell avoids the inclusion of symmetry codes thus simplifying the rules for building the string at the cost of getting a representation that is not very compact (for example a cubic MOF with, say $Z = 48$, even with very few atoms in the asymmetric unit would generate a SLICES with several hundred atoms and a even higher number of edges). This implies repetition of symmetry-related atoms and bonds but redundant information is for sure better than lack of information and nowadays simplicity and reliability of the result are much more important than the length of the string.

Response: Thank you for your positive evaluation and very valuable suggestions. In response, we expanded the description of the SLICES format in the revised manuscript. In addition, we concur with your viewpoint that, “Explicitly encoding all atoms in the unit cell avoids the inclusion of symmetry codes thus simplifying the rules for building the SLICES strings. Although this results in a less compact representation, the trade-off is justified for simplicity and reliability”, and have incorporated this perspective in the revised manuscript.

We greatly value your enlightening suggestion regarding the potential application of SLICES for storing chemical connectivity in MOF databases. Inspired by your input, we evaluated the performance of SLI2Cry on QMOF-21-40 (see Supplementary Note 3 for details). The match rates of 6.19% under loose criteria and 2.95% under strict criteria indicate that the current iteration of SLI2Cry faces challenges for reconstructing MOFs from SLICES strings. To develop an invertible representation for MOFs (termed MOFSLICES), we propose encoding structural building units (SBUs) like organic ligands and metal clusters as single nodes when constructing quotient graphs. The SBU symbols can be represented by their indices in a predefined SBU database. For rebuilding MOFs from their MOFSLICES strings, we can build upon the topology-based MOF construction algorithm proposed by Boyd and Woo³⁶. This hierarchical graph approach that simplifies SBUs into graph nodes could enable MOF reconstruction. The development of MOFSLICES and the reconstruction routine will be a direction for future studies. We have incorporated these discussions into the revised manuscript, and the revisions related to these aspects are outlined in the response to your query: “It would be interesting to see how SLItoCry works with this kind of compounds, maybe testing a set of (say) a few hundred COFs/MOFs with more than 20 atoms in the unit cell.”

Revisions made on page 6 of the main text:

A SLICES string always begins with symbols of atoms in the unit cell (Fig. 1b), encoding the chemical composition of the corresponding crystal structure. ...

... Edge labels, which specify the translational periodicity of edges, are the defining feature of SLICES. They enable the construction of suitable initial guess structures derived from graph theory (Methods). ...

... To disambiguate node indices from edge labels in the string representation, we utilize ‘o’, ‘+’ and ‘-’ to denote ‘0’, ‘1’ and ‘-1’ in edge labels, respectively. This encoding guarantees that ‘0’ and ‘1’ in SLICES refer exclusively to node indices, eliminating potential confusion during model training.

Encoding all atoms within the unit cell in SLICES eliminates the need to incorporate crystal symmetry groups, simplifying the construction rules for SLICES. Although this results in a less compact representation, this trade-off is justified given that state-of-the-art natural language processing (NLP) models excel at handling long sequences, rendering compactness less important. Owing to its simple and clear definition, SLICES could be useful in chemical description and informatics of solid-state materials.

3. It is not clearly told if the software briefly mentioned in "software implementation" (lines 443-447) and in "code availability" (lines 454-459) is for generating the SLICES from experimental crystal structures, for generating theoretical structures from SLICES or both. Presumably, it is for both and, in such case, the pieces of software executing each of the two functions (crystal -> SLICES and SLICES -> crystal) should be clearly differentiated. For example, for descriptive purposes, only the first part would be necessary. This first part is, with no doubt, much easier than the second but perhaps the term "straightforward" (used in line 99) is not fully appropriate and I think that the crystal -> SLICES task deserves some extra detail. I guess that this portion of the software is able to generate the SLICES from crystallographic information (CIF file???) but this is not explicitly told. Also, how is the chemical connectivity generated, just taking the `_geom_bond_*` items provided by the CIF? Calculating all distances between pairs of atoms and deciding which among them represent chemical bonds by defining some threshold? The authors just mention "EconN" and "pymatgen" with no further details.

Response: Thank you for this insightful suggestion. The software mentioned in "Software implementation" is for executing both crystal -> SLICES and SLICES -> crystal. In response, we have added a new subsection "Encoding crystal structures as SLICES strings" to Results to provide more details for the calculation of chemical connectivity.

Revisions made on page 7 of the main text:

Encoding crystal structures as SLICES strings

Many methods have been developed to analyze the connectivity of crystal structures following Pauling's pioneering work in 1929 that introduced the concept of bond strength. For instance, Minimum distance method, Brunner's method³⁰, Hoppe's method of effective coordination numbers (EconNN)³¹ and crystal near-neighbor method³² are well-established methods for identifying near-neighbor environments, and have been implemented in the "local_env" module of the Pymatgen³³ package. Among these methods, EconNN³¹ offers a relatively good compromise between speed, accuracy and robustness to atomic perturbation³⁴. Therefore, analyses of the local chemical environments are performed with EconNN³¹ implemented within Pymatgen³³ to define edges (bonds) for SLICES. Specifically, encoding a crystal structure as a SLICES string involves three steps: (1) Parsing the crystal structure from a file (e.g., Crystallographic Information File³⁵) into a Structure object using Pymatgen³³; (2) Constructing a structure graph based on the Structure

object using the EconNN³¹ algorithm; (3) Extracting the chemical composition, bonding connectivity, and translation vectors from the structure graph to generate the corresponding SLICES string.

4. It is important to say the license under which the software is available (GPL??). Also it should be stated the operating system needed for compilation and execution (GNU/Linux, MS Windows, MAC, any of them?).

Response: Thank you for this valuable suggestion. In response, we have added a description of the license information and operating system requirement to the Software implementation section.

Revisions made on page 30 of the main text:

... It is published under the GNU Lesser General Public License v2.1, which is an open-source license that is recognized by the Open Source Initiative. The operating system needed for compilation and execution is GNU/Linux. ...

5. There is a lot of confusion in the paper between the format and the reconstruction routine, sometimes "SLICES" is used to refer to the format, in other places "SLICES" is used to refer to the rebuilding procedure. Both are not the same thing and I think the acronym "SLICES" should be used exclusively to design just the format and may be some other acronym (I will use "SLItoCry" as example from here onwards) to refer to the rebuilding procedure. The authors should set it more clear when they are talking about one thing or about the other.

For example, line 234: "SLICES was unable to be applied to 10.83% due to ...". This is not true. SLICES (the format) can be applied without problems to all structures, what cannot be applied is SLItoCry.

Also in line 324: "SLICES representation covers the majority of elements ..." is not correct, the representation cover ALL elements of the periodic table. Again, it is SLItoCry or more precisely GNFF, what does not present that full coverage.

Line 26 and line 326: "SLICES reconstructs ...". The format on its own does not reconstruct anything: it is just used as the starting point for the reconstruction.

Line 66 and line 329: "SLICES outperforms past methods ...". SLICES is a format, not a method, the method is SLItoCry.

Response: We apologize for the confusion arising from using the same acronym, SLICES, for both the representation and reconstruction method. We fully agree that using distinct names will improve clarity. Following your enlightening feedback, we have addressed this issue by denoting the reconstruction routine as "SLI2Cry" instead of SLICES.

Revisions made on Abstract:

... The reconstruction routine of SLICES ...

Revisions made on page 4 of the main text:

... The reconstruction routine of SLICES considerably outperforms past methods in accurately rebuilding input crystal structures while maintaining invariances. ...

Revisions made on page 9 of the main text:

... This reconstruction scheme will be denoted as 'SLI2Cry' throughout the remainder of this text. ...

Revisions made on page 12 of the main text:

... It is noteworthy that the SLICES representation can cover all elements of the periodic table. However, the modified GFN-FF potential employed in step (II) limits the applicability of SLI2Cry to crystal structures containing atoms with atomic numbers up to 86. ...

Revisions made on page 13 of the main text:

Table 1 illustrates the reconstruction performance of SLI2Cry for the filtered MP-20 dataset (40,330 crystals): ...

... This highlights the important role of the optimization targeting modified GFN-FF geometry in step (II) to the success of SLI2Cry. ...

Table 1 | Reconstruction performance of SLI2Cry for the filtered MP-20 dataset (40,330 crystals)

Revisions made on page 14 of the main text:

Table 2 | Reconstruction performance for the MP-20 dataset (45,229 crystals) under the loose setting

Method	CDVAE	FTCP	SLI2Cry
Match rate (%)	45.43 ²³	69.89 ²³	84.66

The reconstruction performances of previous methods were evaluated for the MP-20 dataset (45,229 crystals) under the loose setting²³, and their results were compared with SLI2Cry in Table 2. When applied to the 45,229 crystals within the MP-20 dataset, SLI2Cry achieved a match rate of 84.66%. This figure is lower than the 94.95% match rate observed

on the filtered MP-20 dataset comprising 40,330 crystals. This decrease can be attributed to the inapplicability of SLI2Cry to 10.83% of the MP-20 dataset, primarily due to either high atomic numbers exceeding 86 or low dimensionality. ...

... In contrast, SLI2Cry maintains invariances while achieving higher reconstruction performance than FTCP, primarily owing to SLI2Cry's strategy ...

Revisions made on page 23 of the main text:

... SLI2Cry reconstructs input crystal structures in three steps: ...

... SLI2Cry outperforms past methods in reconstructing input structures ...

Revisions made on page 28 of the main text:

... used to optimize the non-barycentric embedding in step (II) of SLI2Cry (Fig. 2).

6. The success rate of SLItoCry in the chosen benchmark (MP-20) is, in fact, impressively high but it must be stressed that the 20 atoms per unit cell limitation is quite restrictive and surely leaves out a lot of structures that would be interesting to be tested, more notably those including organic portions (COFs, MOFs) that are likely to be underrepresented specially if they are highly symmetric, which implies an important bias to the nature of the analysed set. It would be interesting to see how SLItoCry works with this kind of compounds maybe testing a set of (say) a few hundreds COFs/MOFs with more than 20 atoms in the unit cell.

Response: We appreciate this insightful suggestion to benchmark on crystals containing more than 20 atoms per unit cell. Accordingly, we evaluated the reconstruction capability of SLI2Cry on two additional datasets, the filtered MP-21-40 and the filtered QMOF-21-40.

The filtered MP-21-40 comprises 23,560 materials with 21-40 atoms per unit cell from the Materials Project (Methods). The match rates under the loose and strict criteria for the filtered MP-21-40 dataset are 87.88% and 83.73%, respectively (Table S1). Despite a minor performance decrease compared to that of the filtered MP-20 dataset (Table 1), SLI2Cry still achieved high reconstruction fidelity on this more challenging dataset.

The filtered QMOF-21-40 contains 339 MOFs with 21-40 atoms per unit cell from the Quantum MOF database (*Matter* 4.5 (2021): 1578-1597). The match rates of 6.19% under loose criteria and 2.95% under strict criteria for the

filtered QMOF-21-40 dataset indicate that SLI2Cry faces challenges for reconstructing MOFs from SLICES strings (Table S1). This can be primarily attributed to two factors: (1) The barycentric embedding from graph theory, used in SLI2Cry's step (I), might not provide suitable initial guesses for the organic components of MOFs. (2) The rotational degrees of freedom inherent to the organic linkers in MOFs severely hamper structural matching. While SLICES might not be invertible for MOFs, it can still effectively capture and store the chemical connectivity of MOFs.

To address the limitation of SLI2Cry in MOFs reconstruction, we propose MOFSLICES, encoding structural building units (SBUs) like organic ligands and metal clusters in MOFs as single nodes when constructing labeled quotient graphs. The SBU symbols can be represented by their indices in a predefined SBU database. For rebuilding MOFs from their MOFSLICES strings, we can build upon the topology-based MOF construction algorithm proposed by Boyd and Woo (*CrystEngComm* 18.21 (2016): 3777-3792). This hierarchical graph approach that simplifies SBUs into graph nodes greatly streamlines structure reconstruction while avoiding the inclusion of symmetry codes. The development of MOFSLICES and the reconstruction routine will be a direction for future studies. We have added these useful results to the revised manuscript and the Supplementary Information.

Revisions made on page 15 of the main text:

Additionally, we evaluated the performance of SLI2Cry on the filtered MP-21-40, which comprises 23,560 materials with 21-40 atoms per unit cell from the Materials Project (Methods). Despite a minor performance decrease compared to that of the filtered MP-20 dataset (Table 1), SLI2Cry still accomplished high match rates of 87.88% (loose) and 83.73% (strict) on the filtered MP-21-40 (Table S1). Notably, crystals with 1-40 atoms per unit cell account for 77.1% of all entries in Materials Project database, highlighting the broad applicability of SLI2Cry.

We also assessed SLI2Cry on 339 MOFs with 21-40 atoms per unit cell from the Quantum MOF database³⁸ (filtered QMOF-21-40). The match rates of 6.19% under loose criteria and 2.95% under strict criteria indicate the current limitation of SLI2Cry in reconstructing MOFs (Supplementary Note 3).

Revisions made on page 23-24 of the main text:

While SLICES can encode the chemical connectivity of MOFs, SLI2Cry faces challenges for reconstructing MOFs from SLICES strings. To develop an invertible representation for MOFs (termed MOFSLICES), we propose encoding structural building units (SBUs) like organic ligands and metal clusters as single nodes when constructing quotient

graphs. The SBU symbols can be represented by their indices in a predefined SBU database. For rebuilding MOFs from MOFSLICES strings, we can build upon the topology-based MOF construction algorithm proposed by Boyd and Woo³⁷. This hierarchical graph approach that simplifies SBUs into quotient graph nodes could potentially enable MOF reconstruction, which is planned for future work.

Revisions made on page 29 of the main text:

Filtered MP-21-40 dataset

MP-21-40 comprises 24,959 materials with 21-40 atoms per unit cell from the Materials Project database. In MP-21-40, we select materials with formation energy smaller than 2 eV/atom and energy above the hull smaller than 0.08 eV/atom to exclude unstable materials, following Xie *et al.*²³. After excluding crystals containing atoms with atomic numbers beyond 86 and those with low-dimensional structural units, the filtered MP-21-40 dataset consists of 23,560 crystals.

Revisions made on page 4 of the Supplementary Information:

Supplementary Note 3. Reconstruction performance of SLI2Cry for the filtered QMOF-21-40 dataset

QMOF-21-40 contains 928 MOFs with 21-40 atoms per unit cell from the Quantum MOF database³⁸. After excluding MOFs containing atoms with atomic numbers beyond 86 and those with low-dimensional structural motifs, the filtered QMOF-21-40 dataset consists of 339 MOFs. Only 339 MOFs remained in the filtered QMOF-21-40 dataset, primarily owing to a large percentage of MOFs in the database contains low-dimensional components, as identified by the EconNN algorithm.

Table S1 presents the reconstruction performance of SLI2Cry for the filtered QMOF-21-40 dataset. The match rates of 6.19% under loose criteria and 2.95% under strict criteria indicate that the current iteration of SLI2Cry faces challenges for reconstructing MOFs from SLICES strings. This can be primarily attributed to two factors: (1) The barycentric embedding from graph theory, used in SLI2Cry's step (I), might not provide suitable initial guesses for the organic components of MOFs. (2) The rotational degrees of freedom inherent to the organic linkers in MOFs hamper structural matching. While not presently invertible for MOFs, SLICES can still capture and store the chemical connectivity of MOFs.

Revisions made on page 8 of the Supplementary Information:

Supplementary Table 1 | Reconstruction performance of SLI2Cry on the filtered MP-21-40 dataset (23,560 crystals) and the filtered QMOF-21-40 dataset (339 MOFs)

Setting	Match rate (%)	
	Filtered MP-21-40	Filtered QMOF-21-40
Strict	83.73	2.95
Loose	87.88	6.19

7. A minor question is the appearance of many acronyms that have not been defined (CDVAE, GNNFF, FTCP, RNN, ...). They should be defined in parenthesis the first time they appear. Even SMILES and SLICES itself are defined in the abstract but not in the body of the paper (the definition should be repeated in their first appearance in the introduction).

Response:

Thank you for this valuable suggestion. Accordingly, we added the missing definition of acronyms in the revised manuscript.

Revisions made on page 3 of the main text:

... there are several invertible and invariant representations such as simplified molecular-input line-entry system (SMILES)⁸, International Chemical Identifier (InChI)⁹, ...

... Recently, Crystal Diffusion Variational Autoencoder (CDVAE) was proposed by Xie *et al.*²³ to explore the generation of stable materials. ...

Revisions made on page 4 of the main text:

... simplified line-input crystal-encoding system (SLICES). ...

... (2) optimization based on chemically meaningful geometry predicted with modified Geometry Frequency Noncovalent Force Field (GFN-FF)²⁷, ...

Revisions made on page 7 of the main text:

... Hoppe's method of effective coordination numbers (EconNN)³¹ ...

Revisions made on page 11 of the main text:

... Chen and Ong²⁷ developed a universal interatomic potential for materials based on graph neural networks with three-body interactions (M3GNet IAP), ...

Revisions made on page 14 of the main text:

... Fourier-transformed crystal properties (FTCP)²⁰ ...

Revisions made on page 15 of the main text:

... A general recurrent neural network (RNN)³⁹ was trained ...

Revisions made on page 19 of the main text:

... using Atomistic Line Graph Neural Network (ALIGNN)⁴⁵ ...

Revisions made on page 28 of the main text:

GFN-FF is implemented in the semiempirical extended tight-binding (XTB)⁵⁴ package. ...

Reviewer #2 (Remarks to the Author):

the authors developed a framework to inverse design crystals using an invertible crystallographic representation and 3-step structural optimization methods. The representation is a string-based crystallographic representation that satisfies both invertibility and symmetry invariances. The authors showcase the application of this framework to direct narrow-gap semiconductors. This topic is of interest to the material informatics field and the framework showed improvement over past studies. However, I have concerns regarding the metrics used to validate the results.

Response: We appreciate your positive comments and kind suggestions. We improved our work accordingly. Please find below our point-to-point responses (in blue) to your comments (in black). The revisions are shown in blue color in the revised manuscript.

1. In Table 1 and Table 2, the match rate is used as a tool to demonstrate the effectiveness of the 3-step structural optimization process and benchmark the framework with other two studies. It is not clear how the match rate is calculated. How is it defined? How many data points are used as the training data, testing data, and validation data?

Response: We are sorry for causing this confusion. We divided this comment into two parts so that our response can be more clearly understood.

First, the performance of SLICES's reconstruction routine (denoted as SLI2Cry to avoid confusion, per the suggestion of Reviewer #1) is evaluated by the similarity between the reconstructed and original crystal structures. The match rate is defined as the percentage of those structures that meet the matching criteria of Pymatgen's StructureMatcher algorithm. In response, we revised the description of the match rate in the manuscript to make it clear. We also added a description of the StructureMatcher algorithm in Supplementary Note 1.

Second, SLI2Cry is universally applicable across datasets without the need of training. This is attributed to SLI2Cry's rule-based steps (I) and (II) that require no training, along with the pre-trained, transferable interatomic potential employed in step (III). Therefore, all data points in MP-20 dataset (45229 crystals) can be used as testing data. However, the modified GFN-FF potential employed in step (II) of SLI2Cry limits its applicability to crystal structures containing atoms with atomic numbers up to 86. Additionally, Eon's method applied in step (I) of SLI2Cry, is not applicable for low-dimensional (0D, 1D, or 2D) structures. These restrictions leave us with 40,330 crystals of the MP-

20 dataset (89.17%), denoted as the filtered MP-20 dataset. We employed the filtered MP-20 dataset (40,330 crystals) to evaluate the reconstruction performance of SLI2Cry under the loose/strict criteria (Table 1).

The reconstruction performances of CDVAE and FTCP were evaluated in MP-20 dataset (45,229 crystals) under the loose setting. To compare with previous methods, we employed the MP-20 dataset (45,229 crystals) to evaluate the reconstruction performance of SLI2Cry (Table 2). In response, we revised the description of the (filtered) MP-20 dataset to make it clear.

Revisions made on page 12 of the main text:

The reconstruction performance of SLI2Cry is evaluated by the similarity between the reconstructed and original crystal structures. To evaluate the similarity, we utilized the StructureMatcher function of Pymatgen³³ (Supplementary Note 1). The reconstructed and original crystal structures are deemed similar if they satisfy the matching criteria of StructureMatcher. ...

... The loose/strict match rate for a dataset is defined as the percentage of reconstructed crystal structures that meet the corresponding loose/strict matching criteria when compared to the original structures in the dataset. ...

... It contains 45,229 structurally and chemically diverse crystal structures, ...

SLI2Cry is universally applicable across datasets without the need of training. This is attributed to SLI2Cry's rule-based steps (I) and (II) that require no training, along with the pre-trained, transferable interatomic potential employed in step (III). Therefore, all data points in MP-20 dataset can be used as testing data. It is noteworthy that the SLICES representation can cover all elements of the periodic table. However, the modified GFN-FF potential employed in step (II) limits the applicability of SLI2Cry to crystal structures containing atoms with atomic numbers up to 86. ...

Revisions made on page 13 of the main text:

Table 1 illustrates the reconstruction performance of SLI2Cry for the filtered MP-20 dataset (40,330 crystals): ...

Table 1 | Reconstruction performance of SLI2Cry for the filtered MP-20 dataset (40,330 crystals)

Revisions made on page 14 of the main text:

Table 2 | Reconstruction performance for MP-20 dataset (45,229 crystals) under the loose setting

Revisions made on page 2 of the Supplementary Information:

Supplementary Note 1. StructureMatcher algorithm

The StructureMatcher¹ algorithm first reduces the input crystal structures to their primitive cells and rescales them to equivalent volumes. The algorithm then searches for a valid affine mapping between the two cells, within predefined fractional length and angle tolerances. Finally, the maximum root-mean-square displacement between aligned structures normalized by the average free length per atom is computed. If below the site tolerance, the algorithm classifies the structures as similar based on the optimal lattice transformation found via permutation search.

2. Matching rate is a good metric in comparing the reconstruction performance of the generative algorithms. However, aside from the reconstruction performance, generation performance, and property optimization performance are also important metrics for crystal inverse design algorithms. In the FTCP and CDVAE study, the validity rate and success rate are also reported to show the generation performance of the models. It will be interesting to see SLICES's comparison with the other two studies.

Response: Thank you for this very valuable suggestion. We fully agree that it is important to compare the validity rate and success rate of SLICE-based inverse design method with FTCP and CDVAE. We divided this comment into two parts so that our response can be more clearly understood.

First, to compare the generation performance of SLICES-based inverse design scheme with FTCP and CDVAE, we trained an unconditional RNN (termed as ucRNN) on the filtered MP-20 dataset. We evaluated the material generation performance of the SLICES-based ucRNN model using structural and compositional validity metrics proposed by Xie *et al.* (*Bull. Am. Phys. Soc.* 67, (2022).) That is, a structure is deemed valid if the minimal atomic distance exceeds 0.5 Å, and compositional validity requires overall charge neutrality as determined by SMOG (*Journal of Open Source Software* 4.38 (2019): 1361). We sampled 10,000 SLICES strings using the ucRNN model and evaluated the validity metrics on 9,428 reconstructed crystals to evaluate SLICES's generation performance (Table 3). Our method achieves a higher validity than FTCP, while achieving a similar validity as CDVAE.

Second, to compare the property optimization performance of SLICES-based inverse design scheme with FTCP and CDVAE, we trained a conditional RNN (referred to as cRNN) on the filtered MP-20 dataset. We evaluated the property optimization performance of the SLICES-based cRNN model using the success rate proposed by Xie *et al.*

(*Bull. Am. Phys. Soc.* 67, (2022).) That is, the success rate (SR) is defined as the percentage of crystals achieving 5, 10, and 15 percentiles of the formation energy distribution of the training set. We sampled 1000 SLICES strings using the cRNN model and evaluated the SR on 782 reconstructed crystals to assess SLICES 's property optimization performance (Table 3). Our method significantly outperforms CDVAE and FTCP, highlighting the potential of the SLICES representation for inverse design of solid-state materials.

In response, we added a new subsection “Benchmarks on material generation and property optimization” to Results to compare SLICES’s performance with the other two studies. We added a new subsection “ucRNN/cRNN Models for SLICES String Generation” to Methods. Besides, we added Fig. S2 to the Supplementary Information to describe the conditional RNN model for controlled generation of crystals with desired formation energy. We included Table S2 in the Supplementary Information to outline the parameters and training sets utilized in the RNN models.

Revisions made on page 4 of the main text:

... Additionally, SLICES-based inverse design framework significantly outperforms past approaches in generating materials with a desired property. ...

Revisions made on page 21-22 of the main text:

Benchmarks on material generation and property optimization

To compare the generation and property optimization performance of SLICES-based inverse design frameworks with FTCP²⁰ and CDVAE²³, we trained an unconditional RNN (termed as ucRNN) and a conditional RNN (denoted as cRNN) on the filtered MP-20 dataset (Methods and Table S2).

Table 3 | Generation performance and property optimization performance

Method	Generation performance (%)		Property optimization performance (%)		
	Structural validity	Compositional validity	SR5	SR10	SR15
FTCP ^{20,23}	1.55	48.37	2.00	4.00	5.00
CDVAE ²³	100.0	86.70	78.0	86.0	90.0
SLICES	99.72	84.43	97.4	99.2	99.6

We evaluated the material generation performance of the SLICES-based ucRNN model using structural and compositional validity metrics proposed by Xie *et al.*²³ Specifically, a structure is deemed valid if the minimal atomic distance exceeds 0.5 Å, while compositional validity requires overall charge neutrality as determined by Semiconducting Materials from Analogy and Chemical Theory⁴⁹. We sampled 10,000 SLICES strings using the ucRNN model and evaluated the validity metrics on 9,428 reconstructed crystals (Methods). Our method achieves a higher validity than FTCP, while achieving a similar validity as CDVAE (Table 3).

We evaluated the property optimization performance of the SLICES-based cRNN model using the success rate proposed by Xie *et al.*²³ Specifically, the success rate (SR) is defined as the percentage of crystals achieving 5, 10, and 15 percentiles of the formation energy distribution of the training set. The goal of property optimization is to minimize the formation energy per atom for the generated materials. We sampled 1000 SLICES strings using the cRNN model and evaluated the SR on 782 reconstructed crystals (Methods). Our method considerably outperforms CDVAE and FTCP (Table 3), showcasing the potential of SLICES for inverse design of solid-state materials.

Revisions made on page 23 of the main text:

... Moreover, SLICES-based inverse design framework considerably outperforms past approaches in generating materials with a desired property.

Revisions made on page 29-30 of the main text:

ucRNN/cRNN Models for SLICES String Generation

The ucRNN model was trained on the filtered MP-20 dataset (40,330 SLICES). We applied data augmentation to the filtered MP-20 dataset, resulting in 2,009,115 SLICES strings. The RNN architecture applied here is the same with the RNN models used in the inverse design of direct narrow-gap semiconductors (Table S2). The ucRNN was trained for 10 epochs. We sampled 10,000 SLICES strings using the ucRNN model. However, the majority of these SLICES strings contained repeated edges that impeded reconstruction by SLI2Cry, owing to the difficulties of RNNs in learning the complex syntax of long SLICES strings. Using advanced NLP architectures like Transformer⁴⁴ could help address this challenge and is planned for future work. A simple workaround applied in this study was removing all duplicate edges to correct syntax errors, enabling successful reconstruction of 9,428 materials from the 10,000 sampled strings. We then evaluated the validity metrics on these 9,428 generated structures to assess the ucRNN's performance.

The cRNN model was also trained on the filtered MP-20 dataset for controlled generation of crystals with desired formation energy. The model schematic of cRNN for training and generation is given in Fig. S2a. For training, formation energies of crystals in MP-20 were passed as conditions alongside the SLICES string. The architecture of the cRNN model is illustrated in Fig. S2b. To enable conditional generation, we extended the ucRNN with an additional dense layer that transforms the user-specified formation energy into a tensor. The concatenation of this tensor with the embedding tensor of SLICES is fed into a 3-layer stacked gated recurrent unit (GRU). The cRNN was also trained for 10 epochs (Table S2).

For generation, we input a desired formation energy to the model to sample crystals. To generate crystals with minimal formation energy, we sampled 1000 SLICES strings for each of the formation energy targets (-3.0, -4.0, -4.5, -5.0, and -6.0 eV/atom). After removing duplicate edges in sampled strings, we used SLI2Cry to reconstruct the corresponding crystals. The distribution of formation energy (predicted by M3GNet) of reconstructed crystals under these targets are depicted in Fig. S2c. As seen in Fig. S2c, the distribution of formation energy with target = -3.0, -4.0, -4.5 eV/atom is generally centered around the desired value, when taking into account the deviations between M3GNet predictions and PBE calculations. However, setting the target to lower values (-5.0, -6.0 eV) had an adverse impact, owing to the scarcity of training data samples exhibiting formation energies below -4.5 eV/atom (Fig. S2c). In summary, the lowest mean formation energy predicted by M3GNet was achieved using a target of -4.5 eV/atom. Based on this observation, formation energies (at PBE level) of crystals generated with a target of -4.5 eV/atom were used to evaluate the success rate of property optimization.

Revisions made on page 7 of the Supplementary Information:

Supplementary Fig. 2 | Conditional RNN model for controlled generation of crystals with desired formation energy. **a**, Pipeline for training and controlled generation using the conditional RNN model. **b**, The conditional RNN model architecture. **c**, Distribution of formation energy of generated crystals under various user-specified targets ([-3.0, -4.0, -4.5, -5.0, -6.0] eV/atom), compared with the formation energy distribution of the MP-20 dataset. Normal distribution curves are fitted and included for the top six histograms. For the top five histograms, the formation energies were predicted using the M3GNet model. For M3GNet-predicted formation energies, the minimal mean value was obtained with a target of -4.5 eV/atom. For the second histogram from the bottom, the formation energies were calculated using PBE functional.

Revisions made on page 9 of the Supplementary Information:

Supplementary Table 2 | Parameters used in the models

Model	Key parameters	Tasks trained for	Notes
General RNN	Vocabulary size = 96, Embedding dimension = 128, GRU units = 512	Generating crystals as SLICES	Trained on Materials Project crystals with $N_{atom} \in [1, 10]$ and $E_{form} < 0$ for 10 epochs (30,085 SLICES; Augmented dataset: 764,546 SLICES)
Specialized RNN	Vocabulary size = 96, Embedding dimension = 128, GRU units = 512	Generating crystals with direct narrow-gap as SLICES	Trained on direct bandgap semiconductors in Materials Project with $E_g^{PBE} \in [0.1, 0.55]$, $N_{atom} \in [1, 10]$ and $E_{form} < 0$ for 8 epochs (364 SLICES; Augmented dataset: 11,373 SLICES)
Unconditional RNN	Vocabulary size = 106, Embedding dimension = 128, GRU units = 512	Generating crystals as SLICES for evaluating structural validity and compositional validity	Trained on the filtered MP-20 for 10 epochs (40,330 SLICES; Augmented dataset: 2,009,115 SLICES)
Conditional RNN	Vocabulary size = 106, Embedding dimension = 128, Dense layer dimension = 64, GRU units = 512	Generating crystals with desired formation energy as SLICES for evaluating success rate	Trained on the filtered MP-20 for 10 epochs (40,330 SLICES with E_{form} ; Augmented dataset: 2,009,115 SLICES with E_{form})

3. Minor point: In Figure 3, it shows that the generated crystals (sampled from the latent space) passed through a series of filters to be down-selected as the candidates. Will the addition of a property prediction branch to your RNN to shape the latent space make this step more efficient?

Response: Thank you for this enlightening suggestion. Unlike VAE-based methods such as FTCP and CDVAE, which employ machine learning models as encoders, our inverse design routine features a rule-based encoder, eliminating the need for a latent space in our architecture. In our approach, the SLICES representation itself effectively serves as the "latent space". Inspired by your valuable feedback, we trained a conditional RNN (cRNN) model, as mentioned above. The architecture of the cRNN model is illustrated in Fig. S2b. To enable conditional generation, we extended our unconditional RNN model with an additional dense layer that transforms the user-specified formation energy into a tensor. This tensor is then concatenated with the SLICES embedding tensor, and the combined input is fed into a 3-layer stacked gated recurrent unit (GRU). We have described this cRNN model in both the revised manuscript and the Supplementary Information.

Reviewer #3 (Remarks to the Author):

In their manuscript "An invertible, invariant crystallographic representation for inverse design of solid-state materials using generative deep learning" Xiao et al. present a string representation method to describe solid state crystal structures. The authors aim to develop a string representation as successful as SMILES while overcoming its shortcomings, most importantly, inability to represent covalent networks intrinsic to solid state crystal structures. Quotient graphs have been used to analyze such structures before (see for example Gao et al., 2020, doi:10.1038/s41524-020-00409-0), but for me the most interesting part of the manuscript is the employment of a mechanism to invert the representation by reconstructing crystal structures. The authors demonstrate an impressive fidelity of such reconstruction, as well as illustrate the usability of their representation for the design of novel materials.

Response: Thank you very much for your positive evaluation and kind suggestions. We improved our work accordingly. Please find below our point-to-point responses (in blue) to your comments (in black). The revisions are shown in blue color in the revised manuscript.

I have the following comments, questions and suggestions about the manuscript:

1. Why the authors have chosen a string representation with one-hot encoding as input to deep learning? When underlying data are graphs, using them directly as inputs in graph neural networks seems more natural to me. I believe the manuscript could benefit from an explanation of benefits of such choice.

Response: Thank you for this very valuable comment. We agree that graph-based approaches embody a more natural representation of crystal structure. However, string-based methods allow us to leverage the extensive and rapidly evolving field of natural language processing (NLP). For example, state-of-the-art NLP models like Transformer⁴⁴ have shown promising results in *de novo* molecular discovery, as demonstrated by Bagal *et al.*⁴⁷ with MOLGPT, a GPT⁴⁸-style decoder for generating novel molecules with desired properties. Following this analogy, representing crystals as strings could enable GPT decoders to inversely design new solid-state materials, which is planned for future work. This motivated our design choice of a string representation over graph-based approaches in the current study. The SLICES's reconstruction scheme is denoted as SLI2Cry in the revised manuscript to avoid confusion, per the suggestion of Reviewer #1. In response, we have added these discussions to the revised manuscript.

Revisions made on page 6-7 of the main text:

While graph-based representations are more intuitive for crystal structures, string-based representation allows us to take advantage of the extensive and rapidly evolving field of NLP. Based on this consideration, we opted for a string representation over graph-based approaches in this work.

2. The authors demonstrate a high success rate for structure reconstruction from SLICES. However, it would also be interesting to see the analysis of failures, even if just a couple of them.

Response: Thank you for this very valuable suggestion. In response, we analyzed four representative cases where SLI2Cry was unable to reconstruct crystal structures (Fig. S1). We have added these results to the revised manuscript and the Supplementary Information.

Revisions made on page 14 of the main text:

Furthermore, we analyzed four representative cases where SLI2Cry faced challenges in reconstructing original crystal structures (Supplementary Note 2 and Fig. S1). The findings indicate that further improving the accuracy and robustness of modified GFN-FF in step (II) could enhance the performance of SLI2Cry.

Revisions made on page 3 of the Supplementary Information:

Supplementary Note 2. Analysis of unsuccessful structure reconstruction by SLI2Cry

We analyzed four representative cases where SLI2Cry was unable to reconstruct original crystal structures (Fig. S1).

(1) For TbSm_3 (mp-1187379), the rescaled and ZL^* -optimized structure (②, ③) matches the original structure, but the M3GNet IAP optimization on ZL^* -optimized structure (③) encountered an “Exception encountered when calling layer spherical_bessel_with_harmonics” error. (2) For Cu_2O_3 (mp-755040), atomic collisions in the barycentric embedding led to a problematic rescaled structure (②), causing reconstruction failure. (3) $\text{CdPb}_2(\text{ClO})_2$ (mp-1077904) exhibited underestimated bond lengths of ZL^* -optimized structure (③), affecting the reconstruction with M3GNet IAP. (4) For $\text{Sm}(\text{HO})_3$ (mp-625409), the EconNN algorithm overestimated the coordination of certain Hydrogen atoms, resulting in a poor ZL^* -optimized structure (③) and subsequent reconstruction failure. In summary, further improving the accuracy and robustness of modified GFN-FF in step (II) could enhance SLI2Cry's reconstruction performance.

Revisions made on page 6 of the Supplementary Information:

Supplementary Fig. 1 | Analysis of four failure cases of SLI2Cry for crystal structure reconstruction. The original ①, rescaled ②, ZL*-optimized ③, and IAP-refined ④ structures of TbSm₃ (a), Cu₂O₃ (b), CdPb₂(ClO)₂ (c), Sm(HO)₃ (d). The lattice parameters are provided for each structure. Red error marks in the figure represent failed structural refinements using M3GNet-IAP. The original structures are marked in black. Structures that match the original ones are marked in green, while those failing to match the original ones are marked in red.

3. Coming from crystallographic background I find the usage of some terms confusing. First of all, when seeing "symmetry" (for example, line 23) I tend to think about crystal symmetry, but it seems that this term is used in other

sense in most of the text, except probably in line 295. I would suggest explaining the meaning of "symmetry" in more detail. Then in line 324 the authors use term "crystallographic representation" where I think "crystal representation" is more appropriate.

Response: We appreciate your insightful suggestions. We replaced “symmetry-invariances” in abstract with “translational, rotational, and permutational invariances” to avoid the confusion in understanding "symmetry". We also removed all “symmetry” placed before “invariances” in the manuscript. As a result, the “symmetry” in the revised manuscript only refers to the crystal symmetry. In addition, we agree that “crystal representation” is more appropriate than "crystallographic representation". We replaced "crystallographic representation" with "crystal representation" in our revised manuscript.

Revisions made on the title of this work:

An invertible, invariant crystal representation for inverse design of solid-state materials using generative deep learning

Revisions made on page 2 of the main text:

... the lack of an invertible crystal representation that satisfies translational, rotational, and permutational invariances. ...

..., which is a string-based crystal representation that satisfies both invertibility and invariances. ...

... SLICES guarantees invariances. ...

... and invariant crystal representation, ...

Revisions made on page 3 of the main text:

... there are several invertible and invariant representations such as simplified molecular-input line-entry system (SMILES)⁸, ...

..., whereas invariances indicate that representation after rotation, ...

... A representation that satisfies both invertibility and invariances is necessary to enable general and property-driven inverse design using GMs. ...

..., owing to the lack of an invertible, invariant and periodicity-aware crystal representation that covers the majority of elements across the periodic table.

... Crystal graph is an invariant representation ...

Revisions made on page 4 of the main text:

... Utilizing the invariant multi-graph representation, ...

... In short, no work has demonstrated an invertible crystal representation that satisfies full invariances.

... we propose a string-based invertible crystal representation that guarantees invariances, ...

... The reconstruction routine of SLICES considerably outperforms past methods in accurately rebuilding input crystal structures while maintaining invariances. ...

Revisions made on page 7 of the main text:

... to obtain Euclidean embeddings of periodic graph with the maximum acceptable crystal symmetry. ...

Revisions made on page 8 of the main text:

Note that the barycentric embedding is an embedding with maximum acceptable crystal symmetry. ...

Revisions made on page 13 of the main text:

... Given that SLICES maintains invariances ...

Revisions made on page 14 of the main text:

... In contrast, SLI2Cry maintains invariances ...

Revisions made on page 23 of the main text:

We present SLICES, a string-based, invertible and invariant crystal representation. ...

... SLI2Cry outperforms past methods in reconstructing input structures while still preserving invariances. ...

... To our knowledge, SLICES is the first invertible crystal representation that satisfies full invariances. ...

Revisions made on page 24 of the main text:

... and invariant crystal representation, ...

4. Some parts of the text present claims that are not very well based, I would suggest rephrasing them, or removing them altogether. In the abstract (line 31) and introduction (line 70) the authors claim that SLICES has the potential to "become a standard tool", I think it is too early to make such a claim. In line 242 the authors talk about computational efficiency of the reconstruction scheme. In my opinion, a scheme which requires crystal structure reconstruction with forcefields is quite computationally expensive. I believe such claim is appropriate only when comparing reconstruction times with other representations. Also I would suggest rephrasing line 321 to avoid using word "democratize" which is very unclear in this context. Please as well remove words "user-friendly" from line 443, as such claim is inappropriate in primary sources.

Response: Thank you for this very valuable suggestion. To address these issues, (1) We rephrased "become a standard tool" as "shows promise as a useful tool". (2) We removed "SLICES's reconstruction scheme is computationally efficient ". (3) We rephrased "has the potential to democratize" as "showcases potential as a useful tool for". (4) We removed "user-friendly" from the Software implementation section.

Revisions made on page 2 of the main text:

... SLICES shows promise as a useful tool for *in silico* materials discovery.

Revisions made on page 15 of the main text:

... The reconstruction of the filtered MP-20 database (40,330 crystals) was completed within one hour on a workstation with 2 Xeon E5-2699v4 processors (2x22 cores, 2.2 GHz), indicating SLICES is suitable to be integrated into inverse design pipelines of crystals.

Revisions made on page 19 of the main text:

A workstation with dual Xeon E5-2699v4 CPU (2x22 cores, 2.2 GHz) and a NVIDIA RTX 2080 Ti GPU was employed to run the inverse design scheme (Supplementary Note 4). In total, 14 potentially synthetically accessible direct narrow-gap semiconductors with unique compositions and structures were inversely designed in less than 11 days on this workstation.

Revisions made on page 24 of the main text:

... SLICES showcases potential as a useful tool for the inverse design of functional crystalline materials.

Revisions made on page 30 of the main text:

SLICES has been implemented as a Python package. ...

Revisions made on page 5 of the Supplementary Information:

Supplementary Note 4. Sampling speed of SLICES-based inverse design scheme

A workstation with dual Intel Xeon E5-2699v4 CPU (2x22 cores, 2.2 GHz) and a NVIDIA RTX 2080 Ti GPU was employed to run the inverse design scheme. The training of RNN models took ~ 14 hours, while sampling 10 million SLICES strings took ~6 hours. The reconstruction of approximately 3.4 million crystals from SLICES took under 6 days. Additionally, the screening process for identifying promising candidates took around 4 days. In total, 14 potentially synthetically accessible direct narrow-gap semiconductors with unique compositions and structures were inversely designed in less than 11 days on this workstation.

5. I applaud the authors' choice to upload the used software and datasets to FigShare, but I suggest improving provenance and reproducibility of your research. Versions for all pieces of software and datasets have to be indicated. Please cite Git tag or commits for SLICES and the modified XTB package. FigShare uploads also have versions, please cite them as well, because future uploads may cause ambiguity.

Response: Thank you for this valuable suggestion. In response, (1) We specified the Git tag and commit ID for both the SLICES and modified XTB packages in the Software implementation and Code availability sections. (2) We included the FigShare version numbers for the data uploads in the Data availability and Code availability sections.

Revisions made on page 30 of the main text:

... , a Docker image with pre-installed SLICES v1.4 package, modified XTB (commit: 0fcba9e)⁵⁴, ...

Revisions made on page 31 of the main text:

The inverse design data of direct narrow-gap semiconductors and the data for reconstruction, material generation, and property optimization benchmarks can be accessed on Figshare (<https://doi.org/10.6084/m9.figshare.22707472>, Version 2).

The SLICES source code is available on GitHub (<https://github.com/xiaohang007/SLICES>). The SLICES documentation is hosted at <https://xiaohang007.github.io/SLICES/>. SLICES v1.0 was used for the reconstruction benchmark on the MP-20 dataset and inverse design of direct narrow-gap semiconductors. SLICES v1.4 (with no changes applied to SLI2Cry) was used for the reconstruction benchmark on the filtered MP-21-40 and filtered QMOF-21-40 datasets, the material generation and property optimization benchmark. A Docker image containing pre-installed SLICES and dependencies is available on Docker Hub (`docker pull xiaohang07/slices:v3`) and Figshare (<https://doi.org/10.6084/m9.figshare.22707946>, Version 1) to facilitate reproducibility. The modified XTB package (commit: 0fcb9e) can be found at <https://github.com/xiaohang007/xtb>.

6. Certain parts of the results section could benefit from more details. It should be explained what term "augmented" in line 281 means. In lines 290-291 it should be explained why such a decrease happened. When talking about dissimilarity measure in line 299 it would be nice to explain what do lower and higher values mean. Figure 3 could include dataset sizes.

Response: Thank you for this valuable feedback. In response, (1) We expanded the description of data augmentation in the revised manuscript. (2) We have revised the relevant statement in the manuscript to better explain the cause of this decrease. (3) We added an explanation of the structural dissimilarity to the revised manuscript. (4) We have added the dataset sizes to Figure 3.

Revisions made on page 16 of the main text:

Revisions made on page 17 of the main text:

... Arús-Pous *et al.*⁴³ demonstrated that using randomized SMILES improves generative model performance over canonical SMILES. Therefore, we applied SLICES randomization (data augmentation) to both the general dataset (30,085 SLICES) and the transfer dataset (364 SLICES), resulting in 764,546 and 11,373 SLICES strings respectively. The randomization was achieved by arbitrary permutations of atom order and edge order in SLICES strings. ...

Revisions made on page 18 of the main text:

... . Among them, ~3.4 million strings were decoded into crystal structures, while reconstruction was unsuccessful for ~6.6 million strings. This is primarily due to duplicated edges within these strings. This underscores the difficulties of RNNs in learning the complex syntax of long SLICES strings. State-of-the-art NLP architectures like Transformer⁴⁴ could help address this challenge, and is planned for future study. ...

Revisions made on page 18-19 of the main text:

... We evaluated structural uniqueness between designed and training crystals using a dissimilarity value based on site coordination information³². Values near zero signify identical structures, whereas values surpassing 1 represent substantial structural differences. ...

7. Some minor points:

* It is uncommon to start sentences with "And ...", I suggest avoiding such constructions.

Response: Thank you for this valuable suggestion. In response, we rephrased sentences starting with "And ..." in the revised manuscript.

Revisions made on page 3 of the main text:

..., whereas invariances indicate that representation after rotation, ...

Revisions made on page 16 of the main text:

... Moreover, ...

Revisions made on page 28 of the main text:

... Moreover, ...

..., and outputs equilibrium bond lengths/angles ...

* Abbreviation "RNN" (line 252 for example) is not explained anywhere in the text.

Response: Thank you for this suggestion. In response, we have added the definition for "RNN" to the revised manuscript.

Revisions made on page 15 of the main text:

... A general recurrent neural network (RNN)³⁹ was trained ...

* "InChI" is written incorrectly in line 37.

Response: We appreciate the reviewer pointing out this misspelling. In the revised manuscript, we have corrected it to "InChI".

Revisions made on page 3 of the main text:

... International Chemical Identifier (InChI)⁹, ...

* Are the URLs in lines 450 and 452 meant to be identical?

Response: We appreciate the reviewer noticing these duplicate Figshare URLs. The reviewer makes an excellent point that this could lead to reader confusion. To address this issue, we have combined these two sentences into one, and included the version number for the Figshare dataset link.

Revisions made on page 31 of the main text:

The inverse design data of direct narrow-gap semiconductors and the data for reconstruction, material generation, and property optimization benchmarks can be accessed on Figshare (<https://doi.org/10.6084/m9.figshare.22707472>, Version 2).

* Please cite git commit in reference 24.

Response: Thank you for this suggestion. In response, we have cited the git commit in reference 24.

Revisions made on page 34 of the main text:

24. Xie, T. & Fu, X. MP-20 dataset (commit 73874c4). <https://github.com/txie-93/cdvae>.

* Please elaborate references 37 and 44, at least authors and URLs are needed.

Response: Thank you for this suggestion. In response, we have added authors, URLs and commit/version to these two references.

Revisions made on page 36 of the main text:

46. Choudhary, K. & DeCost, B. Pre-trained ALIGNN models (commit c698def). <https://github.com/usnistgov/alignn/> (2023).

Revisions made on page 37 of the main text:

54. Atkinson, P. *et al.* Semiempirical Extended Tight-Binding Program Package v6.6.1. <https://github.com/grimme-lab/xtb> (2023).

* In Table 2, why is the match rate of SLICES different from the one provided in Table 1?

Response: We are sorry for causing this confusion. In response, we revised the explanation of this difference in the manuscript to make it clear.

Revisions made on page 14 of the main text:

... When applied to the 45,229 crystals within the MP-20 dataset, SLI2Cry achieved a match rate of 84.66%. This figure is lower than the 94.95% match rate observed on the filtered MP-20 dataset comprising 40,330 crystals. This decrease can be attributed to the inapplicability of SLI2Cry to 10.83% of the MP-20 dataset, primarily due to either high atomic numbers exceeding 86 or low dimensionality. Nevertheless, the achieved match rate of 84.66% still ...

* "Euclidian" in line 102 should be spelled as "Euclidean".

Response: We appreciate the reviewer pointing out this misspelling. In the revised manuscript, we have corrected it to "Euclidean".

Revisions made on page 7 of the main text:

... to obtain Euclidean embeddings of periodic graph with the maximum acceptable crystal symmetry. ...

* "Systematically" in line 111 should be spelled as "systematic".

Response: Thank you for this suggestion. In response, we have corrected it to "systematic" in the revised manuscript.

Revisions made on page 8 of the main text:

... Eon's method enables systematic optimization of initial guess structures ...

* Generally I find it difficult to understand where figure captions end and the regular text begins.

Response: We apologize for the confusion caused. To address this concern, we have added an empty line below each figure caption to ensure clarity.

* Chemical formulas are not necessary in figure captions of Figures 4 and 5.

Response: Thank you for this suggestion. In response, we have removed the chemical formulas in figure captions of Fig.4 and Fig. 5.

REVIEWER COMMENTS

Reviewer #1 (Remarks to the Author):

In my opinion, the authors have satisfactorily addressed my comments on the first version, hence I consider that the actual version is acceptable for publication.

Reviewer #2 (Remarks to the Author):

The authors made significant efforts to address my concerns. The generation performance of SLICE is validated with the cRNN model. I don't have any further questions.

Reviewer #3 (Remarks to the Author):

The authors have considerably improved their manuscript answering all my previous suggestions and comments. I have the following new observations, mostly related to the newly added material:

1. In line 231 of the manuscript the authors describe the usage of StructureMatcher software, which was not mentioned in previous manuscript version. Please cite version of StructureMatcher used in this study as different versions may produce different matching results.
2. Similarly in line 398 the authors mention the usage of SMOCT software. Please cite its version number as well.
3. "Repeated edges" (line 551 of the main text) could probably be better expressed as "duplicated edges", as is done on line 357 of the main text. If this is the same situation, I think it would be better to use "duplicated" instead of "repeated" in all occurrences.
4. Why different versions of SLICES were used for the reconstruction of MP-20 dataset (v1.0) and filtered MP-21-40 dataset (v1.4) (lines 594-596 of the manuscript)? Why cannot both datasets be analyzed with SLICES v1.4? Is the end user supposed to use different versions of SLICES for different types of materials?
5. In Supplementary Figure 1 steps are numbered as 0, 2, 3 and 4, is this intentional? If yes, such numbering might confuse the reader as there is not step 1. I may assume the step numbers correspond to the manuscript's Figure 2, but such correspondence should probably be indicated explicitly.
6. I personally find cross reference "(Methods)" confusing (see lines 100 and 300 of the manuscript for example, but there might be more occurrences). I think it would be better to write "(see Methods section for details)". Other references like "(Supplementary Note 2 and Fig. S1)" are easier to intuitively understand for me.
7. In line 119 of the manuscript "Minimum distance method" should start with lowercase letter (as "crystal near-neighbor method" does).
8. In line 27 of Supplementary Information "Hydrogen" is written in capital letter. I believe it should start with lowercase letter ("hydrogen").
9. Different apostrophes are used in Xi'an city name in manuscript lines 5, 6, 720 and 722.

Point-by-point response to the reviewers' comments

Reviewer #1 (Remarks to the Author):

In my opinion, the authors have satisfactorily addressed my comments on the first version, hence I consider that the actual version is acceptable for publication.

Response: We would like to express our sincere gratitude for your thorough review of our manuscript and for providing valuable feedback. Your input has been instrumental in the refinement of our paper.

Reviewer #2 (Remarks to the Author):

The authors made significant efforts to address my concerns. The generation performance of SLICE is validated with the cRNN model. I don't have any further questions.

Response: We would like to express our appreciation for your careful review of our manuscript and for your valuable feedback. Your suggestions have played an indispensable role in improving our paper.

Reviewer #3 (Remarks to the Author):

The authors have considerably improved their manuscript answering all my previous suggestions and comments. I have the following new observations, mostly related to the newly added material:

Response: Thank you very much for your positive evaluation and kind suggestions. We improved our work accordingly. Please find below our point-to-point responses (in blue) to your comments (in black). The revisions are shown in blue color in the revised manuscript.

1. In line 231 of the manuscript the authors describe the usage of StructureMatcher software, which was not mentioned in previous manuscript version. Please cite version of StructureMatcher used in this study as different versions may produce different matching results.

Response: Thank you for this valuable suggestion. The StructureMatcher function used in this work was implemented in Pymatgen package v.2022.11.7. In response, we have added the version of Pymatgen package to the revised manuscript.

Revisions made on page 12 of the main text:

..., we utilized the StructureMatcher function of Pymatgen³³ v.2022.11.7 ...

Revisions made on page 2 of the Supplementary Information:

The StructureMatcher algorithm in Pymatgen¹ v.2022.11.7 first reduces ...

2. Similarly in line 398 the authors mention the usage of SMOCT software. Please cite its version number as well.

Response: Thank you for this valuable suggestion. The version number of SMOCT software used in this work is v2.5.2. In response, we have added the version number of SMOCT software to the revised manuscript.

Revisions made on page 21 of the main text:

... as determined by Semiconducting Materials from Analogy and Chemical Theory⁴⁹ v2.5.2. ...

3. "Repeated edges" (line 551 of the main text) could probably be better expressed as "duplicated edges", as is done on line 357 of the main text. If this is the same situation, I think it would be better to use "duplicated" instead of "repeated" in all occurrences.

Response: Thank you for this insightful suggestion. In response, we have replaced “repeated” with “duplicated” in the revised manuscript. We also replaced “duplicate” with “duplicated” in the revised manuscript.

Revisions made on page 29 of the main text:

... the majority of these SLICES strings contained duplicated edges that ...

... A simple workaround applied in this study was removing all duplicated edges to correct syntax errors, ...

Revisions made on page 30 of the main text:

... After removing duplicated edges in sampled strings, ...

4. Why different versions of SLICES were used for the reconstruction of MP-20 dataset (v1.0) and filtered MP-21-40 dataset (v1.4) (lines 594-596 of the manuscript)? Why cannot both datasets be analyzed with SLICES v1.4? Is the end user supposed to use different versions of SLICES for different types of materials?

Response: Thank you for raising this important point. We sincerely apologize for the confusion caused by mentioning different SLICES versions. In the previous version of the revised manuscript, we stated that v1.0 was used for reconstructing the MP-20 dataset. This reference was made because, at the time of conducting this benchmark, SLICES was in its v1.0 version. Subsequently, during the significant revision phase of our work, SLICES underwent an update to v1.4. We fully agree that this description could cause confusions.

It is essential to note that the core reconstruction algorithm of SLICES, SLI2Cry, remained unaltered between versions v1.0 and v1.4. The difference between the two versions lies in the addition of utility functions ("get_canonical_SLICES", "check_SLICES", and "check_structural_validity") and the addition of python scripts for reproducing new benchmarks. We have verified that analyzing MP-20 dataset with SLICES v1.4 yields results identical to those from SLICES v1.0. Therefore, end users need not worry about using different versions of SLICES for various material types. Per your insightful suggestion, we have simplified the statement in code availability section to "SLICES v1.4 was used to generate all results in this work" to avoid misconceptions about version dependencies.

Revisions made on page 31 of the main text:

... SLICES v1.4 was used to generate all results in this work. ...

5. In Supplementary Figure 1 steps are numbered as 0, 2, 3 and 4, is this intentional? If yes, such numbering might confuse the reader as there is not step 1. I may assume the step numbers correspond to the manuscript's Figure 2, but such correspondence should probably be indicated explicitly.

Response: Thank you for this valuable suggestion. Indeed, the numbering scheme in Supplementary Figure 1 was intentionally designed to align with the labeling in the manuscript's Figure 2. We agree that this choice may have raised potential confusions with the absence of step 1. In response, we added a description of such correspondence to the caption of Supplementary Figure 1.

Revisions made on page 6 of the Supplementary Information:

... The numbering scheme for structures in this figure is the same with that of Fig. 2 in the main text to ensure structural correspondence. Structure ①, the barycentric embedding, is not depicted here. ...

6. I personally find cross reference "(Methods)" confusing (see lines 100 and 300 of the manuscript for example, but there might be more occurrences). I think it would be better to write "(see Methods section for details)". Other references like "(Supplementary Note 2 and Fig. S1)" are easier to intuitively understand for me.

Response: Thank you for this valuable suggestion. In response, we have replaced "(Methods)" with "(see Methods section for details)" in the revised manuscript.

Revisions made on page 6 of the main text:

... They enable the construction of suitable initial guess structures derived from graph theory (see Methods section for details). ...

Revisions made on page 15 of the main text:

... materials with 21-40 atoms per unit cell from the Materials Project (see Methods section for details). ...

Revisions made on page 21 of the main text:

... and a conditional RNN (denoted as cRNN) on the filtered MP-20 dataset (see Methods section and Table S2 for details). ...

... and evaluated the validity metrics on 9,428 reconstructed crystals (see Methods section for details). ...

Revisions made on page 22 of the main text:

... SR on 782 reconstructed crystals (see Methods section for details). ...

7. In line 119 of the manuscript "Minimum distance method" should start with lowercase letter (as "crystal near-neighbor method" does).

Response: Thank you for this valuable suggestion. In response, we have replaced “Minimum distance method” with “minimum distance method” in the revised manuscript.

Revisions made on page 7 of the main text:

... For instance, minimum distance method, ...

8. In line 27 of Supplementary Information "Hydrogen" is written in capital letter. I believe it should start with lowercase letter ("hydrogen").

Response: Thank you for this valuable suggestion. In response, we have replaced “Hydrogen” with “hydrogen” in the revised Supplementary Information.

Revisions made on page 3 of the Supplementary Information:

... the EconNN algorithm overestimated the coordination of certain hydrogen atoms, ...

9. Different apostrophes are used in Xi'an city name in manuscript lines 5, 6, 720 and 722.

Response: Thank you for pointing out this discrepancy. In response, we have replaced “Xi'an” with “Xi’an” in the revised manuscript.

Revisions made on page 1 of the main text:

¹School of Chemical Engineering, Northwest University, Xi'an, 710069, China

Revisions made on page 38 of the main text:

School of Chemical Engineering, Northwest University, Xi'an, 710069, China